# QUANTIZING SPACE AND TIME: FUSING TIME SERIES AND IMAGES FOR EARTH OBSERVATION

## ABSTRACT

We propose a task-agnostic framework for multimodal fusion of time series and single timestamp images, enabling cross-modal generation and robust downstream performance. Our approach explores deterministic and learned strategies for time series quantization and then leverages a masked correlation learning objective, aligning discrete image and time series tokens in a unified representation space. Instantiated in the Earth observation domain, the pretrained model generates consistent global temperature profiles from satellite imagery and is validated through counterfactual experiments. Across downstream tasks, our task-agnostic pretraining outperforms task-specific fusion by 6% in $R^2$ and 2% in RMSE on average, and exceeds baseline methods by 50% in $R^2$ and 12% in RMSE. Finally, we analyze gradient sensitivity across modalities, providing insights into model robustness. Code, data, and weights will be released under a permissive license.

## 1 INTRODUCTION

Integrating heterogeneous data modalities is a central challenge in deep learning, particularly when combining spatially rich imagery with temporally dynamic signals. In domains such as climate science, manufacturing, and healthcare, this fusion is critical for capturing complementary information: images provide high-resolution spatial context, while time series encode evolving dynamics. Existing approaches for fusing these different data modalities are largely task-specific instead of task-agnostic, which limit cross-modal interactions and generalization. Recent advances in discrete representation learning have transformed image modeling through tokenization, enabling scalable pretraining and generative modeling. However, analogous approaches for time series remain less explored, and a unified token-based framework for both modalities is missing.

In this work, we introduce a task-agnostic multimodal fusion framework that leverages quantized representations and masked correlation learning to align images and time series in a shared representation space. Our approach supports bidirectional generation of time series from images and vice versa while enabling efficient pretraining for diverse downstream tasks. Our framework is schematically visualized in Fig. 1. We later instantiate our method in the Earth observation domain, demonstrating its ability to generate consistent, unbiased global temperature profiles from satellite imagery and to deliver substantial gains in diverse crop yield prediction tasks across of the US. Finally, we provide interpretability analyses via gradient-based sensitivity, offering insights into the robustness of learned multimodal representations.

Our contribution is three-fold: (i) We explore various strategies for quantizing time series and show the effectiveness of the quantization in cross-modal alignment with quantized image representations, (ii) we provide a path towards generating consistent time series profiles based on imagery at global scale underpinned by counterfactual analyses, (iii) we demonstrate the superiority of our *task-agnostic* cross-modal alignment pretraining over both *task-specific* fusion and a range of baselines in high-value downstream applications.

## 2 RELATED WORK

**Fusing Time Series and Images.** Aiming to combine spatial information from images with temporal dynamics from time series has become a critical direction in multimodal learning. Early

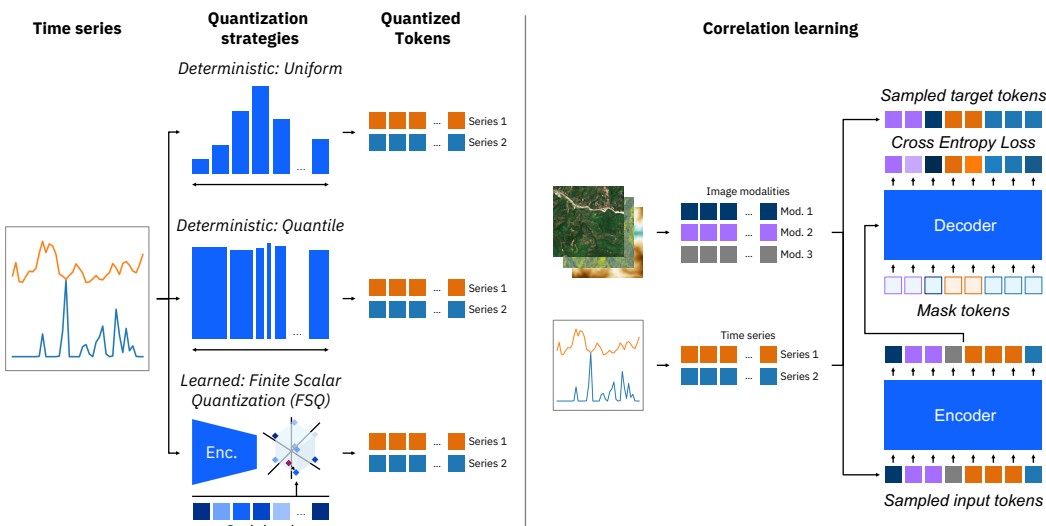

Figure 1: Framework for task-agnostic fusion of quantized time series and image tokens. **Left:** Schematic comparison of tokenization strategies for uni-variate time series. (a) Deterministic: Uniform value-range partitions. (b) Deterministic: Quantile-based partitions for balanced codebook usage. (c) Learned: Encoder–latent finite scalar quantization (FSQ) producing discrete tokens. In addition: Quantization of image patches via learned FSQ. **Right:** Cross-modal alignment learning leverages quantized tokens from images and timeseries to correlate the modalities.

approaches relied on feature concatenation, where CNN-derived image embeddings and RNN-embedded time-series features were merged into a joint representation (Awasthi & Chinzvende, 2025; Wang et al., 2019). In addition, intermediate fusion strategies introduced cross-modal attention mechanisms that allow image tokens to attend to time-series tokens, capturing complementary interactions (Hayat et al., 2022; Liu et al., 2023). In parallel, late fusion approaches combined modality-specific predictions through weighted averaging or gating (Boukhris et al., 2024; Ashfaq et al., 2025). Overall, the above studies, spreading across domains such as healthcare, agriculture, and Earth observation, consistently report that integrating static images with dynamic time series can improve predictive performance, highlighting the complementary nature of spatial context and temporal dynamics. Importantly, we note that existing approaches for fusing time series and images remain supervised and *task-specific*. Instead, in this work, we propose a general, *task-agnostic* fusion approach and summarize high-level differences in Fig 2.

**Quantizing Images.** Discretizing image tokens via quantization has largely been explored through learned discrete vocabularies. Vector-Quantized VAEs (VQ-VAEs) (Van Den Oord et al., 2017) introduced the idea of mapping continuous image features to the nearest entry in a learned codebook, then representing an image as a sequence of discrete indices. Building on this idea, VQGAN (Esser et al., 2021) incorporated adversarial and perceptual losses to improve visual fidelity at higher compression rates, while dVAE produced discrete codes that were adopted in large-scale generative models such as BEiT (Ramesh et al., 2021; Bao et al., 2021). More recently, Finite Scalar Quantization (FSQ) (Mentzer et al., 2023) has been proposed as a simpler and more efficient alternative, discretizing latent dimensions independently to form a combinatorial codebook without the complexities of additional codebook losses.

**Quantizing Time Series.** Discretizing time series was initially explored through classical symbolic methods. For example, piecewise aggregate approximation compresses a series into segment means (Keogh et al., 2001), while symbolic fourier approximation discretizes low-frequency Fourier coefficients (Schäfer & Högqvist, 2012) . Modern approaches often replace hand-crafted quantization with learned tokenizers. VQ-VAEs discretize latent embeddings by mapping them to codebook entries. More recently, TOTEM demonstrated universal tokenization across diverse domains (Talukder et al., 2024), while discrete generative models such as SDformer (Chen et al., 2024) and MSDformer (Chen et al., 2025) showed that multi-scale token sequences can improve long-horizon generation.

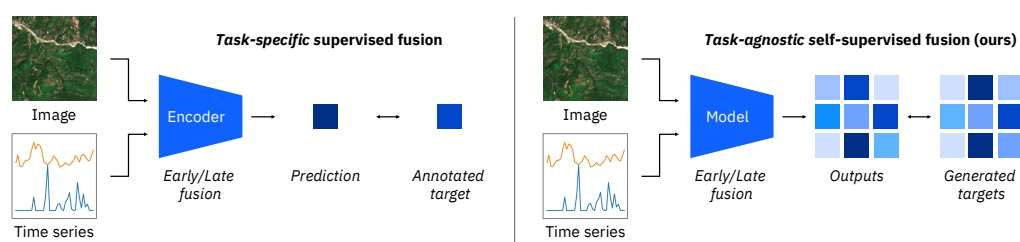

Figure 2: High-level comparison of *task-specific* supervised fusion of time series and images and *task-agnostic* self-supervised fusion (ours).

## 3 METHODS

In the following section, we introduce our methodology for the quantization of time series and images as well as the cross-modal alignment of the resulting tokens. We provide a summary of our approach in Fig. 1. Following the flow of the summary figure from left to right, we below describe: (1) the quantization process for time series, (2) the quantization of images, and (3) the cross-modal alignment of quantized time series and image tokens.

### 3.1 QUANTIZATION OF TIME SERIES

We aim to represent a univariate time series $\mathbf{x} \in \mathbb{R}^L$ of length $L$ as a sequence of quantized tokens $\boldsymbol{\tau} = (\tau_1, \ldots, \tau_{L'})$. In the following, we propose three distinct quantization strategies to achieve this which differently balance the tradeoff between performance and computational effort: Deterministic uniform and quantile quantization map signals to predefined codebook tokens based on binning. In contrast, learned quantization based on FSQ employs a trainable encoder–decoder framework, allowing the tokenization to be adapted during training at the cost of increased computational effort.

#### 3.1.1 DETERMINISTIC QUANTIZATION

**Uniform**. Given global minimum and maximum values $[x_{\min}, x_{\max}]$ of a time series and a predetermined number of codebook tokens $B$, uniform quantization discretizes the range of time series values into uniformly spaced tokens: $\mathcal{T} = \left\{ t_j = x_{\min} + \frac{j}{T}(x_{\max} - x_{\min}) \mid j = 0, \ldots, T \right\}$. Each signal $\mathbf{x}_t$ is then discretized according to: $\tau_t = \arg\max_{j \in \{1, \ldots, T\}} \mathbf{1}[t_{j-1} \leq \mathbf{x}_t < t_j]$. Generally mapping the sequence $\mathbf{x}_1, \ldots, \mathbf{x}_L$ to a sequence of token indices $(\tau_1, \ldots, \tau_{L'})$.

**Quantile**. To address the skewness in the token distributions resulting from uniform quantization of long-tailed distributions, quantile quantization defines boundaries that ensure each token class contains an equal portion of the probability mass. Let $\hat{F}$ denote the empirical cumulative distribution function of the data. For a given number of tokens $T$, the boundaries are set at the empirical quantiles: $\mathcal{T} = \left\{ t_j = \hat{F}^{-1}(j/T) \mid j = 0, \ldots, T \right\}$. This method allocates finer bins in regions of high data density, resulting in a more balanced distribution of token frequencies.

#### 3.1.2 LEARNED QUANTIZATION

We subsequently explore a novel method to quantize time series data based on FSQ. This development has been motivated by initial tests on leveraging VQ-based methods (e.g., Chen et al. (2024)) where we observed failures in convergence due to instabilities in the codebook losses, especially on long-tailed distributions.

**Finite Scalar Quantization**. Let $E_\theta : \mathbb{R}^L \to \mathbb{R}^{L' \times D}$ represent an encoder which transforms an input sequence $\mathbf{x}$ into a latent representation $\mathbf{Z}_e$. We then introduce the discretization of the latent representation via FSQ as follows: For each latent vector $\mathbf{z}_t = (z_{t,1}, \ldots, z_{t,D})$, we determine the index $i_{t,d}$ of the nearest quantization level $c_{d,j}$ from a set of $L_d$ uniformly spaced points within the

range $[-1, 1]$ as $i_{t,d} = \arg\min_{j \in \{0,\ldots,L_d-1\}} |z_{t,d} - c_{d,j}|$, where $c_{d,j} = -1 + \frac{2j}{L_d-1}$. The resulting vector of indices $(i_{t,1}, \ldots, i_{t,D})$ is bijectively mapped to a single token $\tau_t$ from a codebook of size $M = \prod_{d=1}^{D} L_d$ via mixed-radix encoding:

$$\tau_t = \sum_{d=1}^{D} i_{t,d} \prod_{m=d+1}^{D} L_m. \tag{1}$$

The decoder, $D_\phi$, reconstructs the input $\hat{\mathbf{x}}$ from the quantized representation $\mathbf{Z}_q$, which is derived from the token sequence $\{\tau_t\}$. The model is optimized by minimizing a loss function comprising a reconstruction term and a commitment loss. The commitment loss encourages the encoder's output to remain close to the quantization levels:

$$\mathcal{L} = \|\mathbf{x} - \hat{\mathbf{x}}\|_2^2 + \lambda_{\text{FSQ}} \|\mathbf{Z}_e - \text{sg}(\mathbf{Z}_q)\|_2^2, \tag{2}$$

where $\text{sg}[\cdot]$ represents the stop-gradient operator and $\lambda_{\text{FSQ}}$ weights the commitment loss.

### 3.2 QUANTIZATION OF IMAGES

Following recent computer vision approaches, we quantize image data as follows: We define $I \in \mathbb{R}^{H \times W \times C}$ as an image representation. We leverage a vision transformer (ViT)-based encoder, $E_\theta^{\text{quant}}$, which maps non-overlapping patches of the image to a grid of latent vectors $\mathbf{Z}_e \in \mathbb{R}^{h \times w \times D}$. This latent grid is subsequently discretized using FSQ, yielding a quantized representation $\mathbf{Z}_q$ and a corresponding discrete token map $\iota \in \mathbb{Z}^{h \times w}$. A key distinction in our image tokenizer is its generative decoder, $D_\phi$, which is a conditional diffusion model. This model is trained to reverse a noising process. For a given image $I$, a noisy version $I_t$ is generated by adding Gaussian noise $\epsilon \sim \mathcal{N}(0, \mathbf{I})$ at a random timestep $t \in \{1, \ldots, T\}$. The decoder $D_\phi$ of a conditional diffusion model is trained to predict the noise $\epsilon$ that was added to the clean image, given the noisy image $I_t$ at timestep $t$, and conditioned on the quantized latent map $\mathbf{Z}_q$ of the original image. We want to predict the original image based on a noisy input conditioned on the latent from the token The entire architecture is optimized through a composite loss function $\mathcal{L}_{\text{img}} = \mathbb{E}_{I,\epsilon,t} \left[ \|I - D_\phi(I_t, t, \mathbf{Z}_q)\|_2^2 \right] + \lambda_{\text{FSQ}} \|\mathbf{Z}_e - \text{sg}(\mathbf{Z}_q)\|_2^2$, where $D_\phi$ is the denoising network conditioned on the quantized latents $\mathbf{Z}_q$, and $\lambda_{\text{FSQ}}$ is the hyperparameter for the commitment loss.

### 3.3 CROSS-MODAL CORRELATION LEARNING

We propose to leverage an encoder-decoder architecture to learn the correlation between quantized time series and images in a *task-agnostic* way. In this way, we allow the model to co-encode different modalities in the shared encoder and to learn the generation of one modality from the other in the decoder. The transformer encoder, $E_\theta^{\text{corr}}$, processes a concatenated sequence of randomly sampled tokens from both modalities. Its self-attention layers allow tokens to interact across modalities, creating a fused latent representation, $\mathbf{Z} = E_\theta^{\text{corr}}(\mathbf{X})$. Subsequently, the decoder, $D_\phi$, leverages this shared representation $\mathbf{Z}$ through cross-attention to generate the target token sequence $\hat{\mathbf{Y}}_{\text{tgt}}$. The model is trained end-to-end by minimizing the cross-entropy loss between the predicted and ground-truth quantized tokens. Therefore, we leverage the quantized representations similar to target classes in supervised pretraining, expecting different concepts are captured by different tokens of the vocabulary. This task-agnostic objective allows the model to learn stably and establish a powerful joint embedding space that captures the underlying cross-modal relationships.

## 4 APPLICATION TO EARTH OBSERVATION

**Data.** In order to generate a multimodal dataset capturing both time series dynamics and images, we combine two complementary datasets from the Earth observation (EO) and meteorological domains: TerraMesh (Blumenstiel et al., 2025) and NOAA Global Forecast System (GFS) analysis (NOAA Environmental Modeling Center, 2006). TerraMesh provides high-resolution multimodal EO imagery (10 m), while NOAA GFS offers coarser meteorological data at 0.25° in hourly intervals. To align the datasets, we address both spatial and temporal discrepancies. Spatial alignment is performed via (i) mapping each TerraMesh point to the nearest GFS grid cell, and (ii) interpolation by computing weighted averages of surrounding GFS cells (see Fig. 3). Temporal alignment

is achieved by rounding TerraMesh timestamps to hourly and daily resolution and extracting the corresponding GFS time series. We then obtain two aligned datasets as follows: An **hourly dataset**, where for each image, we extract hourly data for 47h before and 12h after satellite image acquisition for variables like surface temperature and precipitation and a **daily dataset**, where for the 29 days prior and 10 days after acquisition, we compute daily mininum, maximum, and mean temperature as well as minimum, maximum, and summed precipitation. The final, merged dataset is released to the community under a permissive license.

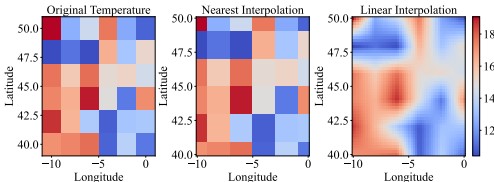 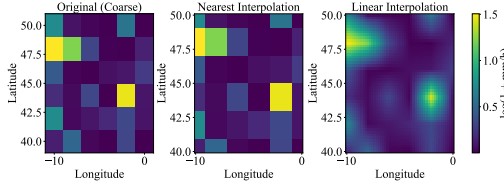

(a) Surface temperature interpolation.        (b) Precipitation interpolation (log-transformed).

Figure 3: Comparison of interpolation strategies for temperature (left) and precipitation (right). Nearest-neighbor interpolation preserves original grid values but introduces blocky artifacts, while linear interpolation produces smoother fields.

**Pretraining Approach.** We warm start our pretraining by leveraging pretrained image tokenizers and backbone weights from TerraMind (Jakubik et al., 2025), a recently released multimodal deep learning model for Earth observation that was pretrained on the TerraMesh dataset we extended above. The backbone has been pretrained on different EO modalities including optical and radar imagery, land-cover maps, vegetation indices, elevation maps, and geo-location tokens. After quantizing both images and time series using the methods described in Sections 3.1 and 3.2, we adopt a continuous pretraining strategy on our merged dataset to align time series and satellite images. At each training step, input and target tokens are sampled independently from Dirichlet distributions. Let $\mathbf{p}^{(\mathcal{I})}$ and $\mathbf{p}^{(\mathcal{T})}$ denote the sampling probabilities for image and time-series tokens, respectively. Then,

$$f\left(\mathbf{p}^{(\mathcal{I})}, \mathbf{p}^{(\mathcal{T})} \mid \boldsymbol{\alpha}^{(\mathcal{I})}, \boldsymbol{\alpha}^{(\mathcal{T})}\right) = \frac{\prod_{i=1}^{M_{\mathcal{I}}} \left(p_i^{(\mathcal{I})}\right)^{\alpha_i^{(\mathcal{I})}-1}}{B\left(\boldsymbol{\alpha}^{(\mathcal{I})}\right)} \cdot \frac{\prod_{j=1}^{M_{\mathcal{T}}} \left(p_j^{(\mathcal{T})}\right)^{\alpha_j^{(\mathcal{T})}-1}}{B\left(\boldsymbol{\alpha}^{(\mathcal{T})}\right)}. \tag{3}$$

where $M_{\mathcal{I}}$ and $M_{\mathcal{T}}$ are the number of image and time-series modalities, respectively, and $\boldsymbol{\alpha}^{(\mathcal{I})}$ and $\boldsymbol{\alpha}^{(\mathcal{T})}$ control their relative weighting. This stochastic sampling ensures that the encoder is exposed to different modality combinations. The sampled tokens are then passed into the encoder–decoder correlation learning framework described in Section 3.3, which learns to reconstruct one modality from the other in a bidirectional, task-agnostic fashion. For time-series generation, the decoder operates in an autoregressive manner, using a causal attention mask so that each token is predicted sequentially conditioned only on its past context.

**Downstream Tasks.** We fine-tune our pre-trained model end-to-end on four downstream tasks in crop yield prediction across the US using the CropNet dataset leveraging both time series and satellite image data (Lin et al., 2024). In this dataset, each county consists of $G$ spatial grids observed over $T$ time steps, where each spatio-temporal sample $(g, t)$ contains a satellite image $I_{g,t}$ and a weather time series $\mathbf{x}_{g,t}^{(\text{weather})}$. Our downstream architecture, illustrated in Fig. 4, leverages the pretrained cross-modal encoder $E_\theta^{\text{corr}}$ to produce cross-modal token embeddings $\mathbf{Z}_{g,t}$ for each sample $(g, t)$. We then apply either apply Mean Pooling (MP) or Modality Mean Pooling (MMP), depending on the experimental setting. For MMP, token embeddings in $\mathbf{Z}_{g,t}$ are first averaged within each modality ($m \in \{\text{image, weather}\}$), and subsequently pooled across all $G$ grids and $T$ time steps. This yields per-modality representations that are concatenated into a joint embedding $\mathbf{z}_{\text{modality}} = \text{Concat}(\bar{\mathbf{z}}^{(\text{image})}, \bar{\mathbf{z}}^{(\text{weather})})$ where $\bar{\mathbf{z}}^{(m)} = \frac{1}{G \cdot T} \sum_{g=1}^{G} \sum_{t=1}^{T} \text{Mean}\left(\{\mathbf{z}_i \in \mathbf{Z}_{g,t} \mid m(\mathbf{z}_i) = m\}\right)$. The fused representation $\mathbf{z}_{\text{modality}}$ is passed to a lightweight MLP prediction head $h_\psi$, trained to minimize the mean squared error $\mathcal{L}_{\text{down}} = \|h_\psi(\mathbf{z}_{\text{modality}}) - y\|_2^2$, where $y$ is the ground-truth yield. For MP, we apply global mean pooling over all token embeddings, ignoring modality boundaries. Finally, after pooling the embeddings, we apply a simple MLP head on the pooled representations to demonstrate the richness of these embeddings in downstream applications.

Figure 4: Downstream task approach for crop yield prediction: Images are encoded individually, followed by modality-wise mean or global mean pooling. Next, spatio-temporal mean pooling merges produced tokens into a single embedding per county. A MLP head predicts overall crop production.

## 5 EXPERIMENTS

In the following, we present experiments on (i) reconstruction performance to understand the effectiveness of different time series quantization approaches, (ii) generation of global surface temperature from optical satellite imagery, (iii) counterfactual analyses to demonstrate the effectiveness of the learned correlation, (iv) downstream performance of our approach on four downstream tasks in crop yield prediction, and (v) the sensitivity of the model with respect to different input modalities.

### 5.1 RECONSTRUCTING QUANTIZED TIME SERIES

In Table 1, we evaluate quantization strategies on hourly global surface temperatures. We observe that the FSQ-based strategy performs best, improving over the deterministic quantile-based method by over 90% in MSE. The FSQ strategy effectively uses less than 300 tokens from the codebook, while deterministic approaches leverage all 500 available tokens. Interestingly, the deterministic uniform approach delivers competitive accuracy on hourly temperatures, overall balancing performance and computational costs. The strong performance likely ties back to the well-defined periodicity of temperature profiles. We provide additional experiments on different quantization strategies and their influence on cross-modal learning in the appendix.

Table 1: Reconstruction performance of different quantization methods for time series.

| Approach | MSE ($\downarrow$) | MAE ($\downarrow$) |
|---|---|---|
| Quantile | 0.0493 | 0.0510 |
| Uniform | 0.0029 | 0.0469 |
| **FSQ** | **0.0023** | **0.0366** |

### 5.2 GENERATING TIME SERIES FROM IMAGES

Our task-agnostic pretraining against quantized token representations not only facilitates cross-modal alignment, but also unlocks cross-modal generation. In the following, we present results from generating daily and hourly global surface temperature from optical Sentinel-2 satellite images. For global-scale experiments, we leverage deterministic uniform quantization to minimize the computational footprint of the experiments. We refer to the appendix for details on other cross-modal generations, for example, conditioning image generation on temperature profiles.

**Daily temperature profiles**. Fig. 5 compares the distributions of generated minimum, mean, and maximum surface temperature across the globe with the actual distribution. Overall, we observe significant overlap between the generated and the actual distributions, with minor differences in the mode as well as slight over-predictions in the areas of 10 to 20°C. Interestingly, the long-tails of the distributions towards low temperatures are accurately captured by the generations. We observe consistent performance for hourly generations, which we discuss in the appendix.

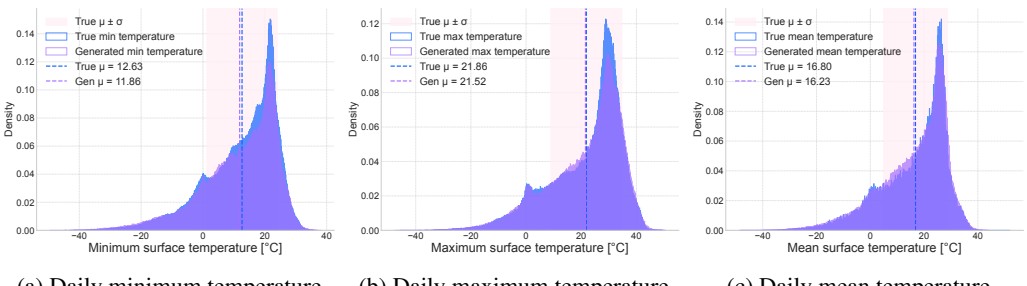

(a) Daily minimum temperature  (b) Daily maximum temperature  (c) Daily mean temperature

Figure 5: Comparison of distribution of true and generated daily surface temperatures across minimum, mean, and maximum temperature profiles.

**Hourly temperature profiles**. Fig. 14 shows generated and actual surface temperature profiles for a random set of geo-locations. Overall, we observe that the generations reflect correct periodicity suggesting the model does not only have a spatial understanding of *where* a satellite image was likely acquired and what temperature is to be expected in this region, but also a temporal understanding of *when* the image was likely acquired, which then determines the position in the temporal periode. This is particularly interesting as the model has neither been trained with a temporal embedding nor leverages information on the geo-location when generating time series, which we will demonstrate in Sec. 5.3. Therefore, our results suggest that the pretrained model is itself able to infer all information that is relevant to generate surface temperature time series only from satellite images.

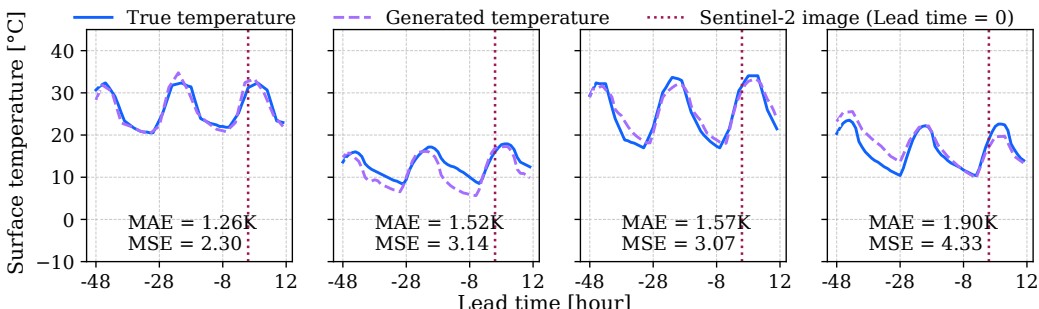

Figure 6: Hourly temperature generations from optical satellite images for a random set of locations.

To better understand the relevance of acquiring latest satellite images for the generation of time series, we measure generation quality across different lead times in Fig. 12 of the appendix. We compare generated and actual surface temperature at the start of the time series (lead time: -47h), at acquisition time of the satellite image (lead time: 0h), and at the end of the timeseries (lead time: +12h). In line with our expectation, we observe the most accurate generations at acquisition time, while especially for low temperature areas, generation quality reduces at -47h and +12h causing slightly worse overall generation performance.

**Global surface temperature.** In Fig. 7, we compare the generated surface temperature at +6h with actual temperatures at global scale. Interestingly, we do not observe spatial biases in the generations. It is worth noting that even fronts, i.e., significant temperature differences between different regions like, for example, Northern India and the Himalaya are well captured in the generations. These results suggest that our pretraining strategy results in a strong cross-modal alignment.

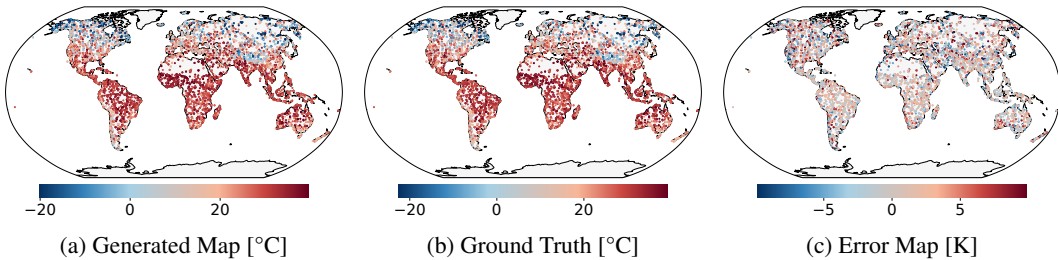

(a) Generated Map [°C]    (b) Ground Truth [°C]    (c) Error Map [K]

Figure 7: Comparison of generated map, ground truth, and error at +6h lead times.

## 5.3 Assessing geo-location sensitivity via counterfactual analysis

To test whether our model learns to generate temperature profiles from visual data rather than exploiting geo-location data as a shortcut—which the model could leverage from the pretrained embeddings of TerraMind (Jakubik et al., 2025)—we design a counterfactual test detailed in Alg. 1. If the model would rely on geo-location data, replacing the true geo-location of an image with a false one would have a significant effect on the generation quality. We provide a visualization for two samples in Fig. 8. The test then measures the change in reconstruction loss between the model-generated and ground-truth temperature series when the true geo-location is replaced by a false one, $\Delta = L_m^{\text{swap}} - L_m^{\text{ref}}$. Our hypothesis is that the model generates temperature profiles solely from image data, leading to $\bar{\Delta} \approx 0$. Using a bootstrap approach avoiding assumptions on the underlying distribution (DiCiccio & Efron, 1996) to test $H_0 : \mu_\Delta = 0$, we find that $\mu_\Delta$ falls within the 95% confidence interval (CI) of $[-0.1580, 4.3095]$. Thus, on average, swapping geo-locations does not produce a systematic change in reconstruction loss, and we find no significant evidence that the model considers geo-location data. We

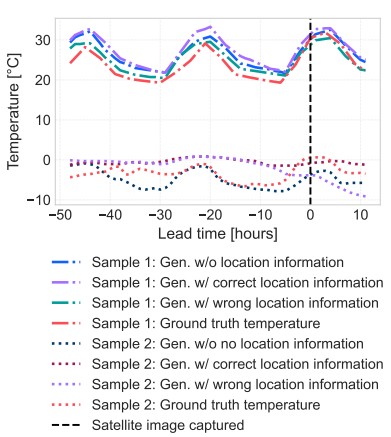

Figure 8: Counterfactual analysis on effect of correct vs. incorrect geo-location on time series generation.

further note that the CI is asymmetric, with a larger positive bound, suggesting that any potential effect of adding geo-location data, if present, would more likely increase reconstruction losses.

## 5.4 Predicting crop yield over the US

Next, we evaluate our pretrained model on four downstream tasks for crop yield prediction comparing against three benchmark models: ConvLSTM (Shi et al., 2015), CNN–RNN (Khaki et al., 2020), and MMST-ViT (Lin et al., 2023). Following the evaluation protocol, all models are trained on data from 2021, model selection is performed based on data from 2022 using $R^2$ scores as the primary criterion Lin et al. (2023). Final results are reported on the year of 2020 as an unseen, hold-out test set. To allow for fair comparisons, we adapt ConvLSTM and CNN–RNN to jointly process satellite imagery and weather time series. Overall, our results in Table 2 demonstrate an average improvement of our *task-agnostic* approach of +6% in $R^2$ (-2% in RMSE) in crop yield prediction accuracy compared to *task-specific* training which translates to a +50% increase in $R^2$ (-12% in RMSE) over baseline methods. The task-specific approach trained from scratch already surpasses the baselines on several tasks. For example, we observe RMSE for Cotton lowered from 199.3 to 184.0, and for Winter Wheat from 18.0 to 12.2. Task-agnostic pretraining further increases these gains: the pretrained versions reduce RMSE for Soybean from 6.1 to 5.7 and for Winter Wheat from 12.2 to 10.5. Finally, our models exhibit substantially smaller validation-test gaps than the baselines. Comparing to Table 5 from the appendix, RMSE for corn increases by only +6% and for Cotton by +3% with our approach, compared to +24% and +8% for baselines, respectively. This stability indicates stronger generalization on unseen hold-out data. Overall, architectural design and pretraining both uniquely contribute to robust and transferable performance gains over existing baselines.

Table 2: Generalization performance measured on unseen holdout year of 2020 as test set.

| Method | Corn | | | Cotton | | | Soybean | | | Winter Wheat | | | Mean Across Tasks | | |
|---|---|---|---|---|---|---|---|---|---|---|---|---|---|---|---|
| | RMSE ↓ | $R^2$ ↑ | Corr. ↑ | RMSE ↓ | $R^2$ ↑ | Corr. ↑ | RMSE ↓ | $R^2$ ↑ | Corr. ↑ | RMSE ↓ | $R^2$ ↑ | Corr. ↑ | RMSE | $R^2$ | Corr. |
| ConvLSTM | 24.0271 | 0.1836 | 0.4285 | 199.2946 | 0.2574 | 0.5074 | 8.7932 | 0.3594 | 0.5995 | 17.9777 | 0.5366 | 0.7325 | 62.52 | 0.334 | 0.567 |
| CNN–RNN | **19.1204** | 0.2999 | 0.5476 | 236.0599 | 0.1231 | 0.3508 | 7.2212 | 0.4547 | 0.6743 | 26.4441 | 0.0783 | 0.2799 | 72.21 | 0.239 | 0.463 |
| MMST-ViT* | 32.7204 | 0.1423 | 0.3772 | 220.6918 | 0.2057 | 0.4535 | 8.4115 | 0.2656 | 0.5153 | 16.6382 | 0.3872 | 0.6223 | 69.62 | 0.250 | 0.492 |
| MMST-ViT** | 25.5541 | 0.1688 | 0.4108 | 281.1944 | 0.1834 | 0.4283 | 8.8070 | 0.3265 | 0.5714 | 15.7938 | 0.3157 | 0.5619 | 82.84 | 0.249 | 0.493 |
| MMST-ViT*** | 28.1470 | 0.1439 | 0.3793 | 222.3744 | 0.1715 | 0.4141 | 6.3668 | 0.3474 | 0.5893 | 11.6998 | 0.3711 | 0.6092 | 67.15 | 0.258 | 0.498 |
| Ours (S \| MP) | 26.4127 | 0.3453 | 0.5877 | 199.4825 | 0.3645 | 0.5465 | 7.2937 | 0.4908 | 0.7006 | 21.4826 | 0.6182 | 0.7828 | 63.67 | 0.455 | 0.654 |
| Ours (S \| MMP) | 20.9062 | 0.3455 | 0.5878 | 184.0204 | 0.4309 | 0.6564 | 6.0794 | 0.4943 | 0.7030 | 12.2222 | **0.6283** | **0.7926** | 55.81 | 0.475 | 0.685 |
| **Ours (P \| MP)** | 21.1918 | 0.3280 | 0.5727 | 196.3347 | **0.4844** | **0.6960** | **5.6936** | 0.5251 | 0.7246 | 14.4440 | 0.5811 | 0.7623 | 59.42 | 0.480 | 0.689 |
| **Ours (P \| MMP)** | 21.9153 | **0.3460** | **0.5883** | **179.6842** | 0.4702 | 0.6857 | 6.3248 | **0.6141** | **0.7837** | **10.5372** | 0.5990 | 0.7740 | **54.62** | **0.507** | **0.708** |

Ablations of our approach: **P**: Pretrained, **S**: From scratch with random weights, **MP**: Mean pooling, **MMP**: Modality mean pooling.
Baseline model sizes: *: tiny, ** : small, ***: medium, following author naming convention (Lin et al., 2023).

## 5.5 MEASURING SENSITIVITY WITH RESPECT TO MULTIMODAL INPUT

In Fig. 9, we assess how sensitive the model is to changes in input modalities by computing modality-specific gradients. In Fig. 9d–9f, we observe a high model sensitivity to changes in the input around the Great Lakes and Southern US, while gradients are significantly lower in the Midwest. Interestingly, this pattern corresponds to the error map for the yield prediction in Fig. 9b. Regions of high error are associated with low gradient magnitudes across all modalities. Thus, in areas the model performs weak, the inputs may provide little useful signal for adjusting the prediction. Second, the gradients behave consistently across modalities, following similar spatial patterns while differing mainly in magnitude rather than direction. Finally, we repeat the experiment after training on all available counties and observe similar error patterns and gradient behavior, further strengthening the hypothesis that in high-error regions the input signals are weak (see appendix for details).

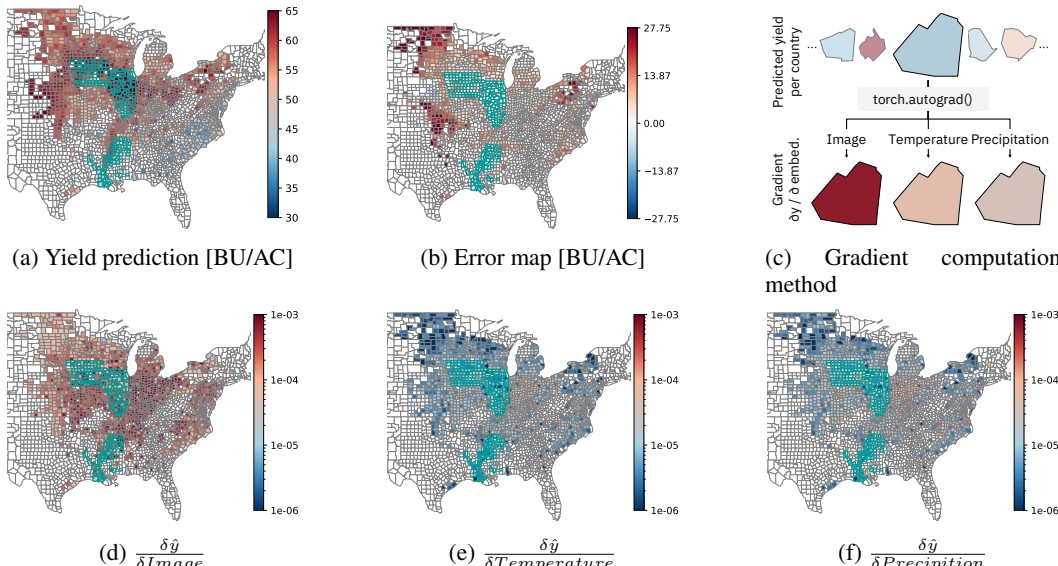

(a) Yield prediction [BU/AC]  (b) Error map [BU/AC]  (c) Gradient computation method

(d) $\frac{\delta \hat{y}}{\delta Image}$  (e) $\frac{\delta \hat{y}}{\delta Temperature}$  (f) $\frac{\delta \hat{y}}{\delta Precipition}$

Figure 9: Gradient analysis: (a) Soybean yield prediction in the US, (b) model errors, (c) gradient generation measuring robustness with respect to changes in input modality embeddings, (d-f) gradients per modality. Counties used for training are highlighted by a green outline (zoom in).

## 6 CONCLUSION

In this work, we demonstrate the relevance of task-agnostic fusion of time-series and images to create information-rich multimodal representations. We provide important insights into time series quantization and cross-modal correlation learning. By instantiating our general-purpose framework in the EO domain, we showcase significant gains over task-specific learning and baselines.

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

APPENDIX

This appendix provides supplementary material and details on our pretraining and downstream fine-tuning, additional results from ablation experiments, and descriptions of relevant algorithms leveraged within the main paper.

## A    PRETRAINING AND DOWNSTREAM ARCHITECTURE DESIGN

In the following, we provide detailed explanations on our pretraining and downstream implementations, including training schedules, hyperparameters, and additional descriptions. We refer to our code for additional explanations which we will release under a permissive license.

### A.1    PRETRAINING DETAILS

**Quantization of Images.** Image modalities are tokenized via pretrained FSQ-based tokenizers further specified in Jakubik et al. (2025). We leverage these tokenizers for Sentinel-1 GRD and RTC products, Sentinel-2 L1C, L2A, and RGB products, NDVI, DEM, and LULC. For satellite images, we use Vision Transformer encoders with patch size 16 and latent dimension $D = 768$. Models are trained with AdamW optimizer, cosine learning rate scheduling, and gradient clipping. Models set $\lambda_{\text{commit}} = 0$ and, therefore, rely on sigmoid regularization. All image modalities besides LULC were tokenized using with a vocabulary size 16K using diffusion decoders. Due to the simplicity of LULC data, the corresponding tokenizer uses a vocabulary with 4K entries and a VAE decoder. Input data is patchified in $16 \times 16$ pixels per patch. geo-location data is treated as text data and use a tokenizer with a 30k vocabulary. Geo-location data is discretized to $0.25°$ resolution and mapped to structured tokens.

**Quantization of Time Series.** As described in the main paper, we explore different approaches to quantization. We generally treated weather modalities as sequence-like data, which allow for both deterministic encoding but also autoregressive decoding. We explore deterministic uniform quantization over the empirical range of each weather variable, deterministic quantile quantization representing adaptive boundaries of bins that adjust to individual distributions in order to transfer skewed raw data distributions into uniform token distributions. Finally, learned finite scalar quantization maps a learned representation into a sequence of four eight-dimensional hypercubes. We employ 1D convolutional encoders with residual blocks and dilated convolutions for capturing multi-scale temporal patterns. Models are trained with AdamW optimizer, cosine learning rate scheduling, and gradient clipping. Time series models use $\lambda_{\text{commit}} = 0.1$. The following special tokens `[PAD]`, `[EOS]`, `[NULL]`, and `[ZERO]` were introduced to handle missing or degenerate values.

**Model Architecture.** We instantiated the backbone of TerraMind (Jakubik et al., 2025) with 12 encoder and 12 decoder layers, with each of them having an hidden dimension of $d = 768$, and 12 heads. Encoder and decoder embeddings are provided per modality. We use SwiGLU-gated MLPs without bias terms in the feed-forward blocks. Normalization is implemented as LayerNorm with $\epsilon = 10^{-6}$. Drop-path rates were set to $0.0$, and all weights were initialized following MAE's initialization scheme. A causal mask is applied only for sequence modalities (e.g., text, weather), while image-like modalities use 2D masking.

The architecture supports *any-to-any generation* and was extended to include quantized weather modalities. For this, we added a discrete weather tokenizer (fixed bins and FSQ variants, see Sec. 3.1). The encoder–decoder setup enables masked token prediction across modalities and autoregressive decoding for sequence modalities.

**Datasets and Loading.** We merged TerraMesh Blumenstiel et al. (2025) with NOAA reanalysis weather data NOAA Environmental Modeling Center (2006) as described within the main paper. Each sample consists of spatially aligned image modalities and interpolated weather sequences. Data were stored in WebDataset shards to allow for efficient data loading. During pretraining, input and target modalities were sampled from a mixture of Dirichlet distributions: (i) Input only modalities, where one untokenized image modality was emphasized ($\alpha = 1000$) and never a target; (ii) balanced mixtures, where tokenized images, coordinates, and weather modalities (e.g., hourly temperature sequences) used $\alpha = 0.5$ for both input and target. Budgets of $N_{\text{in}} = 128$ and $N_{\text{tgt}} = 128$

tokens were allocated by scaling Dirichlet samples, flooring, and clamping to modality-specific limits. Learned positional embeddings were applied to both images (2D) and sequences (1D).

**Distributed Training Setup.** We trained with Distributed Data Parallel (DDP) across four NVIDIA-A100-SXM4 GPUs. Unless otherwise stated, we report the global effective batch size of 128 per GPU (global 8192). Training is controlled by total tokens rather than epochs. With `epoch_size =` 10M and `total_tokens = 10B`, this corresponds to $\sim$4 effective epochs. Pretrain uses bfloat16 mixed precision, learning rates are controlled by a cosine scheduler as follows: Base learning rate of $= 2 \times 10^{-4}$, scaled by `batch_size`/256, yielding an effective LR of $2 \times 10^{-5}$. Cosine decay with warmup over $10^9$ tokens. As an optimizer, we leverage AdamW with $(\beta_1, \beta_2) = (0.9, 0.95)$, $\epsilon = 10^{-8}$, weight decay 0.05. The training is regularized by using gradient clipping at 1.0, with optional skip-update if gradient norm diverged. As stated in the main paper, the model optimizes an modality-averaged cross-entropy loss (`mod`), with additional ordinal EMD loss for weather tokens. Fast dataloading across GPUs is facilitated via WebDataset integration of our raw data with shard-level shuffling at a buffer size of 1000. Each shard is repeated four times for efficient I/O. File descriptor limits were increased to 65k per process to prevent I/O bottlenecks. Model checkpoints are saved every epoch including optimizer and gradient scaler states. Continuous pretraining to incorporate time series data into the EO image model required a compute budget of approximately 4 GPUs for 24 hours, corresponding to $\sim$100 GPU hours on NVIDIA-A100-SXM4.

## A.2 DOWNSTREAM TASKS DETAILS

**Dataset.** For the four downstream tasks using the CropNet dataset, we follow the proposed training and evaluation procedures and train on the year of 2021, validate on 2022, and test on data from 2020. We acknowledge that not all states report crop yields for every crop type and select the following states per crop type to ensure sufficient sample sizes for fair comparisons: **Cotton:** `Alabama, Arizona, Arkansas, California, Florida, Georgia, Kansas, Louisiana, Mississippi, Missouri, New Mexico, North Carolina, Oklahoma, South Carolina, Tennessee, Texas, Virginia`. **Winter Wheat:** `Illinois, Indiana, Kansas, Kentucky, Mississippi`. **Soybean and Corn:** `Illinois, Iowa, Louisiana, Mississippi`.

**Model Architecture.** For the downstream tasks we build on the backbone encoder of our proposed model from the task-specific alignment. We instantiate image and weather encoder embeddings for both temperature and precipitation data and remove the decoder from pretraining. Depending on the experiment, the backbone is either initialized from pretrained weights or trained from scratch with randomly initialized weights. The overall architecture then consists of three major parts: (i) the encoder that processes image patches and weather time series into token embeddings, (ii) a compression module that applies spatial and temporal pooling, and (iii) a lightweight prediction head. We tested two different approaches in the compression module: **Mean pooling** applies spatial and temporal averages across image patches and weather tokens; **Modality mean pooling** separately averages per modality, concatenated pooled embeddings across modalities into a single feature vector. Both strategies produce a sequence-level representation $\mathbf{z}$ of dimension $d = 768$ (or $k \cdot 768$ in the modality-mean case with $k$ modalities). Ultimately, the prediction head is a linear MLP, with residual skip connection: $\hat{y} = \text{MLP}(\mathbf{z}) + 0.1 \cdot W_r \mathbf{z}$., where $W_r$ represents the weight matrix of a residual connection. With the simple decoder, we ensure that performance differences of different encoders can be attributed to the encoder itself, as there is no complex decoder that could perform the heavy-lifting.

**Baselines.** We compare against three benchmark models: ConvLSTM (Shi et al., 2015), CNN–RNN (Khaki et al., 2020), and MMST-ViT (Lin et al., 2023). All models use the same inputs (satellite imagery and weather series) and follow the same protocol: training on 2021, validation on 2022 for model selection, and evaluation on a held-out 2020 test year. To handle multiple geolocations, ConvLSTM and CNN–RNN were extended to jointly process images and weather tensors, treating each grid as an independent sequence and applying mean pooling across time and space. For ConvLSTM and CNN–RNN, we train with MSE loss on standardized log-yields, while MMST-ViT follows the original scheme with MSE on raw log-yields. At each epoch, we report validation RMSE, $R^2$, and Pearson correlation, selecting the best checkpoint by $R^2$.

*ConvLSTM.* A CNN reduces image resolution before sequences are processed by a ConvLSTM. Short-term weather is encoded with an LSTM, projected into additional channels, and concatenated with image features at each time step. Outputs are mean-pooled across time, space, and geolocations, optionally fused with long-term weather embeddings, and passed to a two-layer regressor. Training runs for 200 epochs with AdamW (betas 0.9/0.95, weight decay 0.05), a cosine schedule, and 10 warmup epochs. We use batch size 1 with gradient accumulation of 4.

*CNN–RNN.* Each satellite frame is encoded with a CNN and each short-term weather sequence with an RNN. Visual features are conditioned on weather features via FiLM-style modulation and then pooled across time and space before regression. Training settings match ConvLSTM.

*MMST-ViT.* We follow Lin et al. (2023) by embedding image patches with a PVT backbone conditioned on short-term weather and passed through a Spatial Transformer (across grids) followed by a Temporal Transformer (across time). Long-term weather is projected into context tokens aligned with the temporal dimension. We train tiny, small, and medium variants for 200 epochs with AdamW, a cosine schedule, and 40 warmup epochs.

**Training Setup.** We fine-tuned the model on the CropNet dataset that consists of aligned Sentinel-2 imagery, HRRR weather, and USDA county-level yields. We note that, even for pretrained models, there is a difference in the temporal and spatial resolution of the weather data, requiring to cope with a certain distribution shift. We run the finetuning jobs across 4 GPUs, applied AdamW as an optimizer, learning rate $5 \times 10^{-5}$, weight decay 0.01, batch size 1, and gradient accumulation of 4 steps. The learning rate followed a cosine schedule with 5 warmup epochs. The finetuning process was regularized using dropout at 0.25, attention dropout at 0.3, and stochastic depth at 0.5. We finetuned for 200 epochs and selected the best model based on validation $R^2$ following the proposed procedure.

**Evaluation Metrics.** Following Lin et al. (2023), the performance was generally evaluated on county-level crop yield prediction across the four prediction tasks of soybeans, corn, cotton, winter wheat. As suggested, we report Root Mean Square Error (RMSE) between predicted and observed yields, the coefficient of determination ($R^2$), and Pearson correlation. Metrics were computed on the yield values following USDA statistics. Importantly, we extend the evaluation procedure for additional insights: Instead of reporting performance on the validation year of 2021, we additionally report results on a hold-out, test year of 2020 to better understand the generalizability of different approaches.

### A.3 GRADIENT ANALYSIS DETAILS

To evaluate the model's generalization beyond the training distribution, we compute results and gradients for soybean yield maps across the following U.S. states: `Alabama`, `Arkansas`, `Delaware`, `Georgia`, `Illinois`, `Indiana`, `Iowa`, `Kansas`, `Kentucky`, `Louisiana`, `Maryland`, `Michigan`, `Minnesota`, `Mississippi`, `Missouri`, `Nebraska`, `New Jersey`, `New York`, `North Carolina`, `North Dakota`, `Ohio`, `Oklahoma`, `Pennsylvania`, `South Carolina`, `South Dakota`, `Tennessee`, `Texas`, `Virginia`, and `Wisconsin`.

Among these, `Illinois`, `Iowa`, `Louisiana`, and `Mississippi` are outlined in green to indicate that they were included in the training set.

# B ALGORITHM DESIGN FOR COUNTERFACTUAL ANALYSES

Given images $\mathcal{I} = \{I_m\}_{m=1}^M$, associated coordinates $\mathcal{C} = \{c_m\}_{m=1}^M$, targets $\mathcal{Y} = \{y_m\}_{m=1}^M$, a model $f(I, c)$, and loss $L$, we quantify dependence on coordinates by swapping coordinates and measuring loss inflation. For a subset of $n_{\text{img}}$ images, we compute the reference loss $L_m^{\text{ref}} = L\big(f(I_m, c_m), y_m\big)$. Then, for each image $I_m$, we sample $n_{\text{false}}$ *false* coordinates $c_j$ with $j \neq m$ and compute swapped losses $L_m^{\text{swap}}(j) = L\big(f(I_m, c_j), y_m\big)$. The per-swap deltas $\Delta_{m,j} = L_m^{\text{swap}}(j) - L_m^{\text{ref}}$ are averaged to obtain the image-level sensitivity score $\overline{\Delta}_m = \text{mean}_j \Delta_{m,j}$. Large positive $\overline{\Delta}_m$ indicates that mismatched coordinates substantially degrade performance, i.e., the model relies on coordinates; values near zero suggest coordinate insensitivity. We report $\{\overline{\Delta}_m\}$ and their dataset summary (e.g., mean/median or distribution).

---

**Algorithm 1** Coordinate-Swap Sensitivity Evaluation

---

**Require:** Images $\mathcal{I} = \{I_1, \ldots, I_M\}$; Coordinates $\mathcal{C} = \{c_1, \ldots, c_M\}$; Ground-truth time series $\mathcal{Y} = \{y_1, \ldots, y_M\}$; Model $f(I, c)$; Loss function $L(\hat{y}, y)$; Number of images $n_{\text{img}}$; Number of false coordinates $n_{\text{false}}$.
**Ensure:** Sensitivity scores $\{\overline{\Delta}_m\}$.
 1: Initialize set of processed images $\mathcal{S} \leftarrow \emptyset$
 2: Initialize list of scores $\{\overline{\Delta}_m\} \leftarrow [\,]$
 3: **while** $|\mathcal{S}| < \min(n_{\text{img}}, M)$ **do**
 4:     Sample a new reference index $m \in \{1, \ldots, M\} \setminus \mathcal{S}$
 5:     Add $m$ to $\mathcal{S}$
 6:     Compute model output at true coordinate:

$$\hat{y}(I_m, c_m) \leftarrow f(I_m, c_m)$$

 7:     Compute reference loss:
$$L_m^{\text{ref}} = L(\hat{y}(I_m, c_m), y_m)$$

 8:     Initialize $\mathcal{D}_m \leftarrow [\,]$                                  ▷ per-swap deltas
 9:     **for** $j = 1$ to $n_{\text{false}}$ **do**
10:         Sample a false coordinate $c_j$ with $j \neq m$
11:         Compute model output with swapped coordinate:

$$\hat{y}(I_m, c_j) \leftarrow f(I_m, c_j)$$

12:         Compute swapped loss:

$$L_m^{\text{swap}}(j) = L(\hat{y}(I_m, c_j), y_m)$$

13:         Compute delta:
$$\Delta_{m,j} = L_m^{\text{swap}}(j) - L_m^{\text{ref}}$$

14:         Append $\Delta_{m,j}$ to $\mathcal{D}_m$
15:     **end for**
16:     Compute mean sensitivity for image $I_m$:

$$\overline{\Delta}_m = \text{mean}(\mathcal{D}_m)$$

17:     Append $\overline{\Delta}_m$ to results
18: **end while**
19: **return** $\{\overline{\Delta}_m\}$

---

## C ADDITIONAL EXPERIMENTS AND ABLATIONS

In the following, we provide a range of additional experiments and ablation studies in order to better understand the representations learned as part of our proposed pretraining approach.

**Notation and Metrics.** Unless noted otherwise, temperatures are reported in °C (absolute temperature and statistics) and °K for *errors* or *residuals* (differences). For hourly sequences, $t = 0$ denotes the *Sentinel-2 acquisition time*; a positive (negative) lead time indicates hours after (before) acquisition. For daily sequences, the horizontal axis is in days relative to acquisition. We denote mean absolute error (MAE), mean squared error (MSE), and coefficient of determination ($R^2$) in the captions. NOAA GFS analysis is used as the meteorological reference for surface (2-m) air temperature; Sentinel-2 optical imagery provides the visual context.

### C.1 HOURLY TEMPERATURE GENERATION

In Figure 10, we overlay distributions of actual and generated hourly temperature files from the global scale test set. The overlaid distributions line up closely. We observe that the generated series is slightly warm-biased on average diverging by around 0.5°K from the true mean of 15.6°C. Overall, the distribution comparison indicates the model captures the overall climatological spread of temperatures. This supports the claim of a mostly unbiased global behavior, with a slight positive offset.

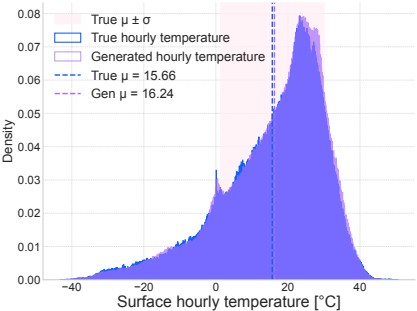

Figure 10: Histogram overlays of true vs. generated hourly surface temperature

We augment the results from Figure 7 by generating global temperature forecasts only using optical Sentinel-2 data as input in Figure 11. We again observe that fronts and regional gradients are preserved and the error map lacks spatial bias which aligns with our findings presented in the main paper. The generations also include the sharp contrasts (e.g., frontal zones) observed before. Overall, these generations are largely consistent with generations obtained for a lead time of +6h, demonstrating the consistency and robustness of the generations.

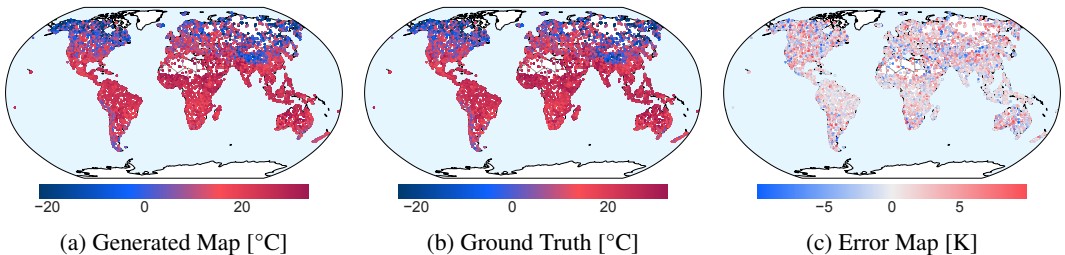

(a) Generated Map [°C]          (b) Ground Truth [°C]          (c) Error Map [K]

Figure 11: Comparison of generated map, ground truth, and error at a lead time of +12h.

In Fig. 12 we compare true hourly surface temperatures from NOAA data with model-generated temperatures on the hold-out test set. We report three representative lead times: the beginning of the time series (-47h), the acquisition time of the satellite image (0h), and the end of the time series (+12h). Across all cases, the scatter plots align closely with the $y = x$ diagonale reference

line, with coefficients of determination consistently above $R^2 = 0.90$. Moreover, no systematic bias is observed across temperature ranges, indicating that the model produces reliable temperature estimates irrespective of the forecast horizon. We observe the lowest errors accompanied by the highest $R^2$-scores for a lead time of 0h, which aligns with our expectation. Generally, highest errors are observed for generations for lead times that are the most far apart from the acquisition of the satellite image, which again is in line with our expectation. In these cases, for example, for a lead time of -47h, the generations exhibit a slightly larger spread around the diagonal, especially in low temperature regimes.

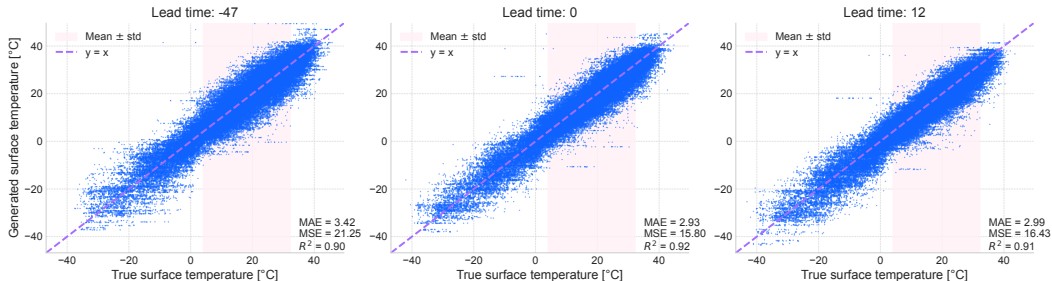

Figure 12: Comparison of true and generated surface temperatures at three representative lead times: the beginning of the sequence $(-47\text{h})$, the image acquisition time $(0\text{h})$, and the end of the sequence $(+12\text{h})$. The shaded region denotes mean $\pm$ standard deviation.

To further illustrate model generating capabilities on the unseen, hold-out set, Fig. 13 shows the parity plot for a lead time of +6h together with the corresponding residual distribution. The residuals appear unbiased and centered around zero, supporting the conclusion that generation quality remains stable across samples from different regions across the globe characterized by diverse temperature profiles.

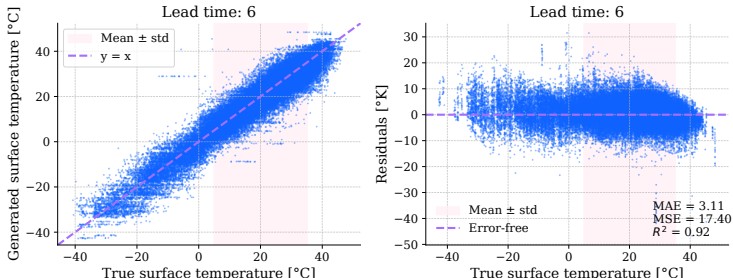

Figure 13: Parity plot of true versus generated surface temperatures at lead time +6h (left) with the corresponding residual distribution (right).

## C.2 DAILY TEMPERATURE GENERATION

In Fig 14, we assess the generated *daily* temperature profiles for randomly selected locations. We not only generate mean, but also minimum and maximum daily temperatures in one shot just using optical Sentinel-2 satellite images as input. We observe that our model has a descent understanding on the overall relationship of means, minimum, and maximum temperatures. We also acknowledge that this approach is not designed or intended to be used as a forecast, as the model more generates averages rather than it could infer the weather ten days ahead just from an optical satellite image, which is obvious but worth mentioning. Therefore, this experiment as well as others are meant to demonstrate the strong correlations that are learned between time series and images from our task-agnostic pretraining.

In Table 3, we compare errors between across generations for mean, minimum, and maximum surface temperature. Overall, we observe that while $R^2$-scores remain comparable and high, the MSE

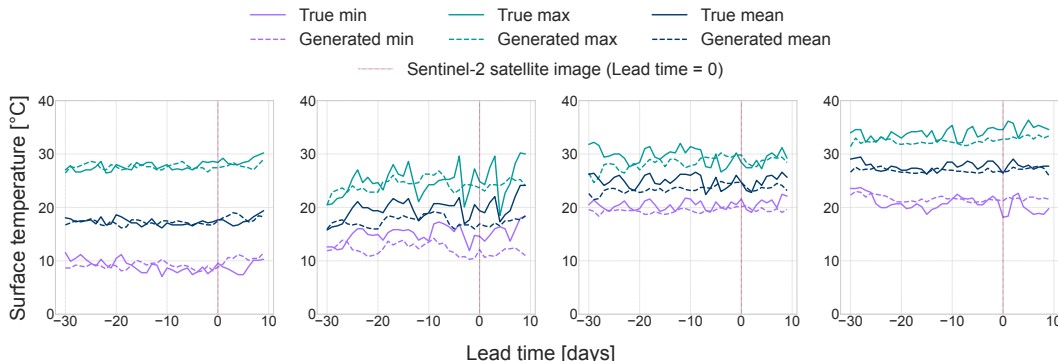

Figure 14: Daily minimum, maximum, and mean surface temperature time series generated from Sentinel-2 imagery around the acquisition day (t=0). Generated curves closely track ground truth across lead times.

and MAE scores are particularly high for maximum temperature generations. Therefore, we observe that while all generated variables appear to have similar correlations with the ground truth, maximum temperature generations exhibit higher errors.

Table 3: Performance of our model in generating daily minimum, maximum, and mean surface temperature time series from Sentinel-2 imagery. Results were generated using the deterministic uniform quantization strategy.

| Variable | MSE ($\downarrow$) | MAE ($\downarrow$) | $R^2$ ($\uparrow$) |
|---|---|---|---|
| Minimum surface temperature | 20.5140 | 3.1721 | 0.8569 |
| Maximum surface temperature | 25.5186 | 3.6955 | 0.8533 |
| Mean surface temperature | 20.7314 | 3.1852 | 0.8637 |

### C.3 RECONSTRUCTION OF QUANTIZATION TIME SERIES

In order to better understand the influence of the number of utilized tokens from the codebook compared to the reconstruction performance, we measure associated errors and $R^2$-scores in Table 4. Overall, we observe that for implementations using 128 tokens or less, the reconstructions perform already reasonable in $R^2$-scores, yet the error can be reduced dramatically when using a larger codebook with more than 300 tokens actively used by the model. We acknowledge that the general codebook utilization for reconstruction hourly temperatures is on the low end for these experiments ranging between 10% and 50% which is expected based on the simple periodicity of surface temperature profiles.

Table 4: Reconstruction performance of Finite Scale Quantization across different codebook settings.

| Token Utilization | Code Dim. | Levels | MSE ($\downarrow$) | MAE ($\downarrow$) | $R^2$ ($\uparrow$) |
|---|---|---|---|---|---|
| 42 | 3 | 8-8-8 | 0.0102 | 0.0778 | 0.9999 |
| 87 | 4 | 4-4-4-4 | 0.0129 | 0.0861 | 0.9999 |
| 128 | 5 | 3-3-3-3-3 | 0.0111 | 0.0792 | 0.9999 |
| 313 | 4 | 8-8-8-8 | **0.0023** | **0.0366** | **0.9999** |

For the specific example of quantizing hourly temperature profiles, we further investigate the distribution of the tokens used by individual approaches. In Figure 15, we see that the token distribution resulting from deterministic uniform quantization matches the actual distribution of the raw data accurately, which is obvious as this approach effectively represents a fixed binning with 500 individual bins. Quantile binning results in a much more uniform token distribtion, aligning with our

expectation. Finally, we see that the distribution of learned tokens differs from both the ones for uniform and quantile quantization heavily. This is not only true for the shape of the distribution but also for the magnitude of individual histogram bins. This suggests that learned quantization follows a different approach, breaking the timeseries into concepts, which are then combined together to form a timeseries. Some concepts appear much more frequently than others, reflected in the largely different distributions.

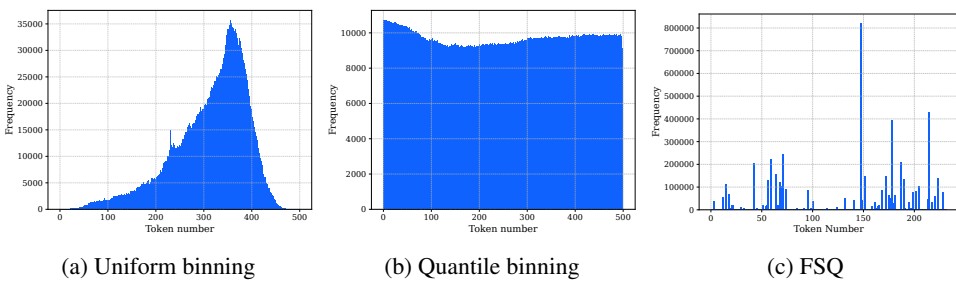

(a) Uniform binning          (b) Quantile binning          (c) FSQ

Figure 15: Token distribution per method.

### C.4 IMAGE GENERATION FROM TIME SERIES

Our model generally facilitate the generation from any modality to any other modality. Therefore, we further explore the influence of time series data on the generation of satellite images. Therefore, we start from the generations of the original TerraMind model by (Jakubik et al., 2025). We specifically generate optical Sentinel-2 data from land use (LULC) and radar Sentinel-1 RTC (S1RTC) data. We observe that the out-of-the-box optical generations by Terramind itself exhibit limited quality, which is in line with the results from the original TerraMind paper (Jakubik et al., 2025). We then specifically condition the generations on different time series to see if the model responds to variations in the temperature magnitudes.

Based on conditioning the generations of optical satellite images additionally on temperature profiles of different magnitudes, the model reacts by first moving to greener vegatation, ultimately become more brown reflecting brush lands. In the first scenario, we provide a temperature profile of a mean temperature of 14.6°C, that we then increases with the second scenario to a mean temperature of 23.9°C. We observe that the response to this change is reflected by additional green artefacts in the generated image. By then increasing the temperature to a mean of 33.4°C in the third and a mean of 42.0°C in the fourth scenario, we observe the generated images respond by more brown, brush-like generations. Overall, the results suggest that the out-of-the-box generations of optical data of the original TerraMind model are limited in quality, however, we do see a response of the continuously pretrained model we propose on different temperature profiles, which underlines that the representation learning itself has effect. We acknowledge the generation quality of TerraMind itself for optical satellite images is of poor quality in this case.

### C.5 DOWNSTREAM APPLICATION

As outlined in Section 5.4 and following the proposed evaluation procedure in Lin et al. (2023), we leverage data from the year of 2022 as the validated split. In Table 5, we report Root Mean Square Error, $R^2$ and Correlation performance on the this validation set of 2022 for Corn, Cotton, Soybean, and Winter Wheat. Across crops, our pretrained modality mean pooling achieves up to a 25% reduction in RMSE compared to task-specific models (e.g., 174.0 vs. 233.0 for Cotton) and boosts correlation by more than 10 points (0.8871 vs. 0.7735 for Winter Wheat). Even for Corn, where ConvLSTM reaches the lowest RMSE (19.3), our pretrained variants yield the highest $R^2$ (0.7150) and correlation (0.8456). For Soybean, pretrained models improve $R^2$ from 0.5984 (ConvLSTM) to 0.7074, corresponding to a relative gain of 18%. These improvements are consistent across metrics, suggesting that pretraining enriches cross-modal embeddings in a way that benefits both accuracy and robustness. Interestingly, when comparing the performance on the validation and test datasets, we observe that our proposed models performs much more stable on fully unseen data compared to baseline approaches.

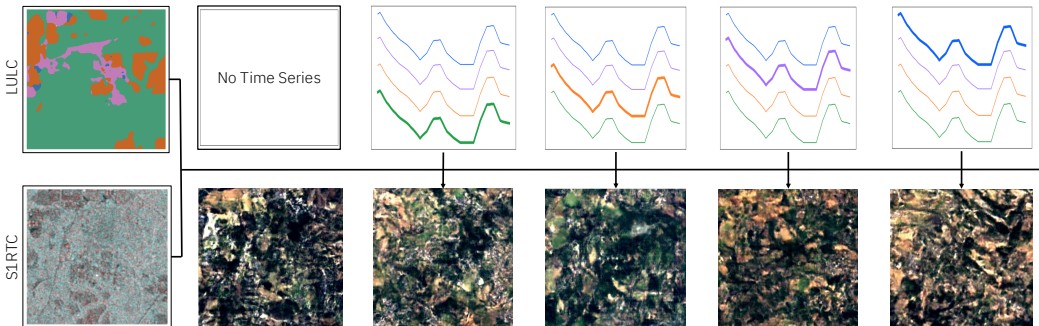

Figure 16: Satellite image generations conditioned on different temperature profiles. Outputs are generally conditioned on LULC and Sentinel1-RTC modalities. Output without timeseries represents generation capabilities of TerraMind (Jakubik et al., 2025) for optical satellite data. Subsequent generations show the influence of gradually increasing temperature data.

Table 5: Downstream performance on the 2022 validation set across four major crops. Best results per crop and metric are highlighted in bold, second best are underlined.

| Method | Corn | | | Cotton | | | Soybean | | | Winter Wheat | | |
|---|---|---|---|---|---|---|---|---|---|---|---|---|
| | RMSE ↓ | $R^2$ ↑ | Corr. ↑ | RMSE ↓ | $R^2$ ↑ | Corr. ↑ | RMSE ↓ | $R^2$ ↑ | Corr. ↑ | RMSE ↓ | $R^2$ ↑ | Corr. ↑ |
| ConvLSTM | **19.3095** | 0.6617 | 0.8134 | 178.6494 | 0.4005 | 0.6329 | 7.1953 | 0.5984 | 0.7735 | 13.1063 | 0.7780 | 0.8821 |
| CNN–RNN | 23.7309 | 0.6504 | 0.8065 | 217.8898 | 0.2698 | 0.5194 | 8.0143 | 0.5707 | 0.7554 | 25.3512 | 0.4909 | 0.7007 |
| MMST-VIT (Tiny-version) | 20.9123 | 0.6369 | 0.7981 | 211.5451 | 0.2758 | 0.5252 | 7.7730 | 0.4803 | 0.6930 | 13.1847 | 0.6427 | 0.8017 |
| MMST-VIT (Small-version) | 19.2542 | 0.6450 | 0.8031 | 233.0100 | 0.3015 | 0.5491 | 7.7564 | 0.5499 | 0.7416 | 15.8803 | 0.4695 | 0.6852 |
| MMST-VIT (Medium-version) | 22.3131 | 0.6184 | 0.7864 | 211.1415 | 0.2877 | 0.5364 | 6.9212 | 0.5553 | 0.7452 | 13.3762 | 0.5754 | 0.7585 |
| Ours (Random mean pooling) | 24.7597 | 0.6071 | 0.6071 | 258.6529 | 0.1982 | 0.4450 | 7.0653 | 0.6233 | 0.7895 | 17.2167 | 0.7359 | 0.8578 |
| Ours (Random modality mean pooling) | 24.2078 | 0.6131 | 0.7830 | 193.2485 | 0.4542 | 0.6739 | **6.0306** | 0.6493 | 0.8058 | **10.4206** | 0.7542 | 0.8684 |
| Ours (Pretrained mean pooling) | 21.3858 | **0.7150** | **0.8456** | 250.7735 | 0.3606 | 0.6005 | 6.2405 | **0.7074** | **0.8411** | 12.4772 | 0.7783 | 0.8822 |
| Ours (Pretrained modality mean pooling) | 20.6163 | 0.7045 | 0.8394 | **174.0135** | **0.4682** | **0.6843** | 6.1207 | 0.6933 | 0.8326 | 10.4622 | **0.7869** | **0.8871** |

## C.6 CAUSALITY EVALUATION

We conducted a coordinate-swap counterfactual test to assess whether the model additionally relies on geo-location metadata rather than solely on satellite image data. For each of $n = 200$ satellite images, we generated an hourly temperature time series conditioned on the true paired coordinates and computed the mean squared error (MSE) against NOAA reference data. The same image was then re-evaluated with $K = 50$ randomly sampled coordinates, producing swapped losses $L_{\text{false}}^{(k)}$. For each image, we computed the mean increase in loss, $\bar{\Delta}_m = \frac{1}{K} \sum_{k=1}^{K} \left( L_{\text{false}}^{(k)} - L_{\text{true}} \right)$. Averaging across all images yielded $\mu_\Delta = \frac{1}{n} \sum_{i=1}^{n} \bar{\Delta}_i$. To test the null hypothesis $H_0 : \mu_\Delta = 0$, we employed a non-parametric bootstrap procedure, which avoids distributional assumptions. Table 6 reports the resulting 95% confidence interval. Since the interval contains zero, we are not rejecting $H_0$. This indicates that, on average, swapping geo-locations does not produce a systematic change in reconstruction loss, and we find no significant evidence that the model leverages geo-location metadata in this setting. In addition, we observe that the CI is asymmetric, with a larger positive bound, suggesting that any potential effect of adding geo-location data, if present, would more likely increase reconstruction losses.

Table 6: Bootstrap test of mean sensitivity delta $\Delta$ under coordinate-swap perturbation. We report the 95% confidence interval (CI) for the mean difference in reconstruction loss. Since the CI includes zero, we fail to reject the null hypothesis ($H_0 : \mu = 0$).

| Statistic | Value |
|---|---|
| Bootstrap 95% CI | $[-0.1580, 4.3095]$ |
| Hypothesis Test | Cannot reject $H_0$ |
| Conclusion | No significant evidence that $\bar{\Delta}$ is different from zero. |

## C.7 Continuous pretraining

In Figure 17, we compare model convergence behavior and the magnitudes of training losses during correlation learning of quantized time series and quantized images. As stated in the main paper, we expect that different approaches of quantizing time series might influence how the model learns to correlate time series and images. Putting it differently: Each quantization approach results in different token representations in terms of distributions and codebook sizes—we expect that these differences have different effects on learning a correlation to a corresponding satellite image.

We generally observe, that the overall magnitude of the correlation learning loss is heavily influenced by the size of the codebook and, thereby, by the possible "classes" the model has to learn to choose from. For example, when the model is trained on time series tokens that were quantized with FSQ and a codebook of 4K, the model has to learn to predict one of 4,096 classes for a time series token leveraging input e.g., a satellite image. That is more challenging than if a model has to learn to predict one of 243 classes if FSQ used a smaller codebook to quantize the data. Therefore, we observe vastly different levels of losses. In line with our expectation, uniform and quantile quantization perform in the middle based on a codebook size of 500.

It is worth noting that a smaller codebook is beneficial from a correlation learning perspective as the complexity of the data is reduced. However, obviously, the expressiveness of tokens from a smaller codebook tend to be smaller than of tokens from a larger codebook. Therefore, we acknowledge that there is a general trade-off in codebook size between expressiveness and the efficiency in correlation learning. We are eager to see exciting work on this in the future.

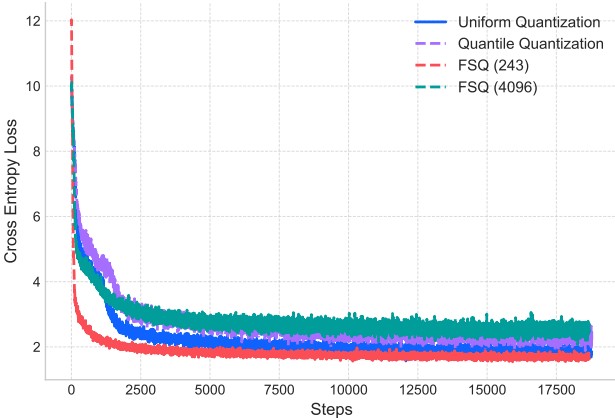

Figure 17: Cross-entropy training loss curves during cross-modal correlation learning based on time series tokens produced by different quantization approaches represented by uniform, quantile, or FSQ tokenizers (243 and 4096 tokens). All settings converge stably under identical training budgets; see Appendix A for optimizer and schedule details.

## C.8 Gradient analysis plot

In Figure 18, we repeat the experiment described in Section 5.5, but this time *training on all available* states listed in Appendix A.3. The resulting patterns closely mirror those observed in Figure 9. Specifically, Figures 18c–18e show gradient distributions that are highly comparable to those in Figures 9d–9f. In both cases, regions with low gradient magnitudes align with regions of high prediction error, further supporting the hypothesis that in areas of high errors, the input signals themselves are weak. We also tested the gradients of the model on the training data, observing a rather uniform distribution of gradients across counties.

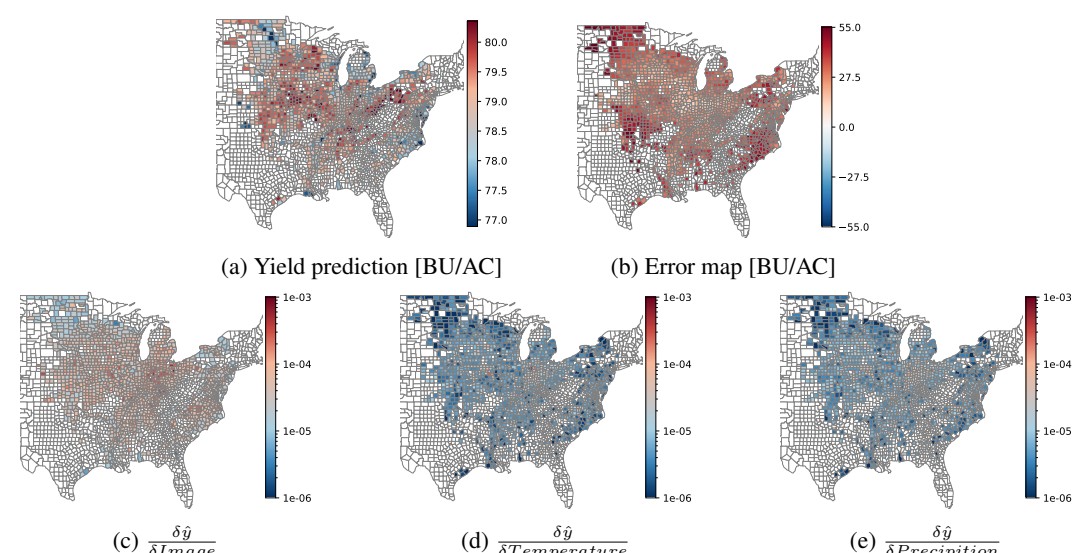

(a) Yield prediction [BU/AC]        (b) Error map [BU/AC]

(c) $\frac{\delta \hat{y}}{\delta Image}$          (d) $\frac{\delta \hat{y}}{\delta Temperature}$          (e) $\frac{\delta \hat{y}}{\delta Precipition}$

Figure 18: Gradient analysis for unseen, hold-out data from 2020 generated for a model trained on all counties based on data from the year of 2021.

## D   LARGE LANGUAGE MODEL ASSISTANCE

We used generative artificial intelligence tools for grammar correction and stylistic refinement during the writing of this paper. The models were not used for content generation or ideation. Additionally, we employed large language models to support the literature review.

