# OpenReview forum: "Quantizing Space and Time: Fusing Time Series and Images for Earth Observation"
_ICLR.cc/2026/Conference — ICLR 2026 Conference Desk Rejected Submission_

### Official Review · Reviewer_Saqg · 2025-10-27

**Soundness:** 3
**Presentation:** 3
**Contribution:** 2
**Rating:** 4
**Confidence:** 3

**Summary:**

This paper proposes a framework for fusing time series data with single timestamp images that relies on quantization of the time series and image representations. In particular, the authors propose exploring a finite scalar quantization method for the time series.  The authors address the following limitations in existing approaches for fusing time series and images: they are supervised and task-specific. Instead, the correlation between time series and images is learned through encoding both modalities in a shared encoder and  learning to generate one modality from the other. The model is pre-trained on a dataset of EO imagery, TerraMesh alongside meteorological data from the NOAA Global Forecast System.
The pre-trained model is fine tuned on four downstream tasks of crop production

**Strengths:**

The paper is generally well written and clear.
The authors explore various methods for quantizing time.  provides some interesting analyses of the method, in particular the assessment of geo-location sensitivity.
The problem of fusing time series information with single timestamp image data is interesting and tackling it can be useful for many applications across agriculture, climate and biodiversity.

**Weaknesses:**

My main concern is about the claim that this model is task-agnostic.
The authors try the method on 4 downstream tasks but they all come from the same dataset, so it's essentially one task of crop yield prediction.
Also there is not necessarily a big gain of performance of the model trained from scratch vs with the pre-trained weights. Therefore, I am wondering whether this claim that the model is task agnostic might be misleading. In the sense that it is an architecture that can be used for different problems , but this is probably also the case for the methods that the authors quality as task-specific.
Have the authors tried transfer learning instead of fine-tuning? it might make more of a compelling case for the task-agnosticity of the model if the authors do not try their method on another evaluation dataset.

As stated by the authors in C.2 for daily temperature generation: "We also acknowledge that this approach is not designed or intended to be used as a forecast, as the model more generates averages rather than it could infer the weather ten days ahead just from an optical satellite image"
So if the pretext task is not of any practical relevance, it would have been good to showcase the potential of the method on another dataset on top of CropNet. If no other dataset is available to try this method, can the authors provide more concrete applications of their model?

**Questions:**

Besides answering to the point brought up in the weaknesses, I have the following questions:
- Can you specify the computational cost of FSQ vs the other methods?
- Can you specify details on how you split the dataset for pre-training and the evaluation of 5.2?
- In L239-240, you specify that the model is pre-trained with different EO modalities and geolocation tokens. Is the geolocation pixel wise or there is only one location associated with each image and in that case, how are the geo-location tokens  integrated with the raster data?
- Do you have an idea of how consistent the times at which the satellite imagery was taken? What is the distribution of timestamps of the image data?
- Fig. 7 : Could you put the error map in celsius as well?
- 5.4: why is the test set in the past (2020)? It is usually more common with crop yield prediction tasks to do forecasting. Also in  MMST-ViT, the task that is considered is one year ahead prediction. Would it make more sense to train on 2020, validate on 2021, evaluate on 2022? The setup that you choose is actually not the one proposed by CropNet originally. Could you clarify why you made this choice for evaluation?
- Have the authors tried other designs for the hourly and daily datasets ? What was the rationale behind the choice of time ranges?

I am open to revising my score if the authors address my concerns. Thank you.

---

> ### Author Response · Authors · 2025-11-19
>
> We thank the reviewer for the feedback and for recognizing the potential broad applicability of our work. Please find our detailed responses to each of your concerns below.
>
> ***Comment Part 1***
>
> > Can you specify the computational cost of FSQ vs the other methods?
>
> Regarding the computational cost of the different quantization methods, this is a crucial point. We are happy to provide a detailed breakdown.
>
> The methods we evaluated fall into two distinct categories with significantly different computational profiles:
>
> 1.  **Deterministic Methods (Uniform and Quantile):** These approaches are **parameter-free** and have negligible computational cost. They operate by calculating bin boundaries directly from data statistics (e.g., min/max values or quantiles). This involves **no training phase**, and at inference time, the quantization is a simple, near-instantaneous mapping operation.
>
> 2.  **Learned Method (FSQ):** In contrast, Finite Scalar Quantization (FSQ) requires training a dedicated encoder-decoder architecture (as described in Section 3.1.2). This introduces a tangible cost during the pre-training phase.
>
> To quantify this, we measured the resources required to train the FSQ model used in our experiments:
>
> *   **Training Duration:** 6 hours
> *   **Hardware:** 4 $\times$ NVIDIA A100 (40GB)
> *   **Total Compute:** 24 GPU-hours
> *   **Energy Consumption:** Approximately **10.8 kWh** (assuming a node power draw of ~1.8 kW).
> *   **Carbon Footprint:** Due to our infrastructure's location in Switzerland (carbon intensity approx. 12-15 gCO$_2$ /kWh), the total emissions were approximately **0.15 kg CO$_2$**.
>
> > Can you specify details on how you split the dataset for pre-training and the evaluation of 5.2?
>
> The dataset used for pre-train and evaluate our framework was described in TerraMesh: A Planetary Mosaic of Multimodal Earth Observation Data [1], the dataset is the combination of two datasets MajorTOM-Core and SSL4EO-S12. Looking at the paper they perform a random 99%-1% train-validation split to the selected MajorTOMCore grid cells, in addition any SSL4EO-S12 patch that intersect the validation extent is added to the validation set; further patches spatially overlapping the joined validation are are removed to mitigate data leakage.
>
> > In L239-240, you specify that the model is pre-trained with different EO modalities and geolocation tokens. Is the geolocation pixel wise or there is only one location associated with each image and in that case, how are the geo-location tokens integrated with the raster data?
>
> For geolocations, we round the latitude and longitude values from the center of each patch to the nearest quarter degree and save the resulting discretized coordinates as strings in a well-defined format (e.g., “lat=35.75 lon=-89.00”) [2]
>
> > Do you have an idea of how consistent the times at which the satellite imagery was taken? What is the distribution of timestamps of the image data?
>
> For a given geographical sample in TerraMesh dataset, the different modalities are closely aligned in time. For Sentinel 2 the revisited time is 5 days, and for sentinel 1 can be 5 days as well, but it depends on the type of sensor. On another note, In TerraMesh, over 80% of the data is single timestamp, meaning that we do not leverage different images from the same spatial area for the majority of the dataset [1]. The rest of TerraMesh (i.e. the multitemporal part) is only seasonal data, therefore, we can rule out that the model can learn simple correlations in temperature generation based on similar revisiting times.
>
> > Fig. 7 : Could you put the error map in celsius as well?
>
> Definitely. In this case, the errors are identical between Kelvin and Celcius, but we are happy to change the legend to Celcius if preferred.
>
> ***Reference***
> 1. Blumenstiel, Benedikt, et al. "Terramesh: A planetary mosaic of multimodal earth observation data." Proceedings of the Computer Vision and Pattern Recognition Conference. 2025.
> 2. Jakubik, Johannes, et al. "Terramind: Large-scale generative multimodality for earth observation." arXiv preprint arXiv:2504.11171 (2025).

---

> > ### Author Response · Authors · 2025-11-19
> >
> > ***Comment Part 2***
> >
> > > 5.4: why is the test set in the past (2020)? It is usually more common with crop yield prediction tasks to do forecasting. Also in MMST-ViT, the task that is considered is one year ahead prediction. Would it make more sense to train on 2020, validate on 2021, evaluate on 2022? The setup that you choose is actually not the one proposed by CropNet originally. Could you clarify why you made this choice for evaluation?
> >
> > We generally follow the CropNet benchmark approach. From their two settings of one year ahead prediction and predicting the crop yield at the end of the season, we decide for the latter as it is the more informative task. We take as input satellite Sentinel-2 imagery and daily HRRR data during the growing season for predicting the yield at the end of the year.
> >
> > One shortcoming of the CropNet benchmark setting is the absence of an individual test set that is treated as a hold out.
> > In this paper, we introduce this test set as a separate year. Given the setting we are working in (predicting in the same year starting from the input of that year), we assume that each year is independent from the previous year, allowing us to interchangeably select individual years as train, validation and hold-out test set. We decided to train on the same year the CropNet benchmark is trained on for full comparability and test on a separate holdout year as a more comprehensive assessment of the actual generalizability. Therefore, we had to take 2020 as test set, as there is no data from 2023 or later available. With these adjustments, we hope to improve these minor benchmark shortcomings of CropNet.
> >
> > > Have the authors tried other designs for the hourly and daily datasets ? What was the rationale behind the choice of time ranges?
> >
> > The time ranges were not arbitrary but were chosen to provide the model with essential temporal context around the point of satellite image acquisition. The period before the image capture (e.g., -47 hours) provides the necessary antecedent weather conditions that directly impact what is observed in the image. The shorter period after capture (e.g., +12 hours) is included to help the model learn a more robust, bidirectional relationship between the visual snapshot and its surrounding temporal dynamics. While other ranges are possible, we established this as a reasonable and broadly relevant baseline for our task-agnostic framework. We will clarify this motivation in the final paper.
> >
> > > There is not necessarily a big gain of performance of the model trained from scratch vs with the pre-trained weights.
> >
> >  Thanks for this remark. Given the use of identical architectures, theoretically both models could converge to an optimal set of weights. In the past, we observed extensive work demonstrating the acceleration from pretraining. Besides this acceleration, our experiments further demonstrate accuracy gains across settings.
> >
> > >  Have the authors tried transfer learning instead of fine-tuning?
> >
> > Thank you for this excellent question, which gives us the opportunity to clarify a critical detail of our experimental setup. In our work, we employ **end-to-end fine-tuning**, which is a standard and effective form of transfer learning where all model weights are updated for the downstream task. Our primary motivation for this choice was not just to achieve maximum performance, but was in fact a **principled necessity due to a significant domain shift** between the pre-training and downstream data. Specifically, our model was pre-trained on standardized Sentinel-2 imagery, which preserves the original radiometric range. In contrast, the CropNet dataset used for our downstream task provides Sentinel-2 imagery that has been compressed to an **8-bit integer format**. This difference in data distribution, value range, and quantization fundamentally alters the input statistics. Due to this domain shift, a feature extraction approach (i.e., freezing the pre-trained backbone and only training a new prediction head) would be suboptimal. The initial layers of the frozen model would be processing data from a distribution they were not trained on, leading to misaligned and likely ineffective features. Therefore, end-to-end fine-tuning was essential to allow the entire network to adapt to this new data domain, from the input layers through to the final prediction head.

---

> > > ### Author Response · Authors · 2025-11-19
> > >
> > > ***Comment Part 3***
> > >
> > > > It would have been good to showcase the potential of the method on another dataset on top of CropNet
> > >
> > > We appreciate this valid point. Our focus on CropNet was necessitated by the scarcity of public benchmarks combining co-aligned satellite imagery and weather time series. To our knowledge, CropNet is the only large-scale dataset currently available for this specific multimodal fusion task.
> > > Nevertheless, CropNet provides a rigorous evaluation via four distinct prediction tasks (corn, cotton, soybean, wheat) across diverse geographies. Our consistent improvements across all tasks demonstrate the robustness of our approach. We view our framework as a foundational step that effectively demonstrates the value of this fusion, hopefully catalyzing the development of future multimodal datasets.

---

> ### Comment · Reviewer_Saqg · 2025-11-25
>
> Dear authors,
> Thank you for responding to my questions.
>
> I would like to ask additional questions regarding your answer about the split chosen for the downstream tasks on CropNet.
> I am still very confused by your splitting strategy. Reading the CropNet paper, this is what I see regarding the crop yield prediction task: "We conduct experiments on the CropNet dataset for 2022 crop yield predictions by using satellite
> images and daily weather conditions during growing seasons, as well as monthly meteorological
> conditions from 2017 to 2021"
> So in your case do you use data from 2017-2021 excluding 2020?
> Or maybe I am not understanding your setup (because the paper states you follow the evaluation from CropNet, but in cyour rebuttal, you wrote "Given the setting we are working in (predicting in the same year starting from the input of that year)," )
> Would you mind clarifying this?
>
> With this confusion still in mind, I would support that it makes more sense to train on 2020, validate on 2021, (then you could train again on 2020+2021) and test on 2022. But maybe I'm missing something, in which case I'd be happy if you can provide some clarifications. Alternatively, as you mention your setup "allow[s] [you] to interchangeably select individual years as train, validation and hold-out test set.", it might be worth doing the different rotations of year/split assignment and averaging because a given year might be more easy than another one.
>
> I agree with your response on the scratch vs with the pre-trained weights. However in that case, it would be good to mention for how many epochs the best models in each setting are trained (is there actual acceleration). It is a bit hard to assess whether the pre-trained model is useful at all for such tasks otherwise.
>
> Thank you for clarifying the difference in preprocessing for your pre-training data vs the CropNet dataset. It would be valuable to add that to the paper. That being said, given that there is much effort put into the pre-training phase, it would be interesting to see if this model can transfer to some tasks to some extent...

---

> > ### Author Response · Authors · 2025-11-25
> >
> > We sincerely thank the reviewer for this follow-up question. We see exactly where the confusion lies, and we appreciate the chance to clarify the difference between our input setup and the original CropNet paper's setup.
> >
> >  **Clarification on Inputs and Data Leakage:**
> > The reviewer is correct that the original CropNet paper uses a historical context vector ("monthly meteorological conditions from 2017 to 2021") as an input feature. However, in our specific setting, we do not use this multi-year historical vector.
> > Our model inputs are strictly limited to the satellite imagery and daily weather conditions of the current growing season only. For example:
> > - To predict the 2020 yield, the model only sees data from the 2020 growing season.
> > - To predict the 2021 yield, the model only sees data from the 2021 growing season.
> >
> > Because we exclude the historical variables used in the original CropNet paper, there is no data leakage. The 2021 training samples contain no information regarding 2020. This effectively decouples the years, allowing us to treat them as independent distributions.
> >
> > We hope this clarifies that our experimental design maintains strict independence between the training (2021) and test (2020) years.

---

### Official Review · Reviewer_85Xn · 2025-10-30

**Soundness:** 2
**Presentation:** 2
**Contribution:** 2
**Rating:** 4
**Confidence:** 4

**Summary:**

The paper introduces a task-agnostic method for multimodal fusion between time series data and single-timestamp images with applications in teh domain of satellite images. It employs quantization strategies for time series and uses a masked correlation learning objective to align discrete tokens from both modalities within a shared representation space. Applied to Earth observation, the pretrained model generates coherent global temperature profiles during pretraining and enhances crop yield prediction downstream task.

**Strengths:**

- Fusing static images with timeseries is important in the domain of satellite images as it combines high-resolution spatial context with time series encode evolving dynamics.
- Literature and methods around feature quantization are explored in detail.
- Code and Data will be provided by the authors.

**Weaknesses:**

- The modeling architecture is not exposed in detail, in particular the architecture and size, how the model understands to separate images and time series inputs are separated and how/if position is explicitly encoded. The proposed method is a pretraining method but it is compared againsta existing architectures. The baseline performance of the model in table 2 should be added to the baselines and results should be gathered for competing models as well to ensure a clear comparison.
- The motivation for employing quantization is not clear (see questions). Alternatives are not explored.
- The method is presented as agnostic but performance is evaluated in a single task (yield prediction).
- Establishing model performance across several pretraining sets (ideally combinations) and downstream tasks is important to assess the generalization of the method.

**Questions:**

- What is the motivation behind the choice of quantizing model inputs? Does it act as a means of corrupting and reconstructing inputs? Have you considered alternative ways of achieveing this?

---

> ### Author Response · Authors · 2025-11-19
>
> We thank the reviewer for the feedback and for recognizing the importance of fusing static images with timeseries. Please find our detailed responses to each of your concerns below.
>
> ***Comment Part 1***
>
> > The modeling architecture is not exposed in detail, in particular the architecture and size, how the model understands to separate images and time series inputs are separated and how/if position is explicitly encoded. The proposed method is a pretraining method but it is compared againsta existing architectures.
>
> We utilize a standard Transformer Encoder-Decoder architecture (based on the TerraMind backbone [1]).
> - **Separation of Modalities**: The model distinguishes between images and time series via modality embeddings that are added to the tokens of each respective input. This allows the shared encoder to process them in a unified sequence while retaining modality identity.
> - **Positional Encoding**: We use distinct positional encodings for each modality: **2D sine-cosine embeddings** for image patches (preserving spatial structure) and **1D positional embeddings** for time series tokens (preserving temporal order).
> - **Size**: As detailed in Appendix A.1, the model uses a ViT-B equivalent size (12 layers, 768 hidden dimension).
>
> To ensure clarity, we will integrate these critical architectural details directly into the main methodology section of the final manuscript.
>
> > The baseline performance of the model in table 2 should be added to the baselines and results should be gathered for competing models as well to ensure a clear comparison.
>
> Thanks for the feedback and thanks for questioning to isolate the benefit of the architecture from the benefit of pretraining. We would like to clarify that **this baseline performance is already included in Table 2.**
>
> - The rows labeled **'Ours (S | MP)'** and **'Ours (S | MMP)'** represent our model trained **from Scratch (S)** with random initialization.
> The rows labeled **'Ours (P | ...)'** represent the Pretrained (P) version.
> By comparing **'Ours (S)'** against the external baselines, we demonstrate that our architecture alone is highly competitive. By comparing **'Ours (P)'** against **'Ours (S)'**, we explicitly quantify the additional gain provided by our task-agnostic pretraining. We will make the **'S vs. P'** notation more explicit in the caption of Table 2 to ensure this comparison is clear.
>
> > The motivation for employing quantization is not clear (see questions). Alternatives are not explored.
>
> Our primary goal was to create a unified representation space where heterogeneous modalities, spatially dense satellite imagery and temporal weather sequences, could be correlated effectively (**representation learning**). We considered the alternative of regressing continuous time series values directly from image embeddings (using MSE loss). However, our approach is informed by findings from recent large-scale multimodal systems like **4M** [2] and **TerraMind** [1]. These works demonstrated that directly regressing continuous values across diverse modalities often leads to training instabilities and balancing difficulties.
> Inspired by this, we made the principled decision to discretize the time series. This transforms the cross-modal fusion problem from a regression task (which can be unstable) into a classification task (predicting discrete tokens). This allows us to leverage a highly scalable masked modeling objective with a standard cross-entropy loss, ensuring robust convergence.
>
> **Reference**
> 1. Jakubik, Johannes, et al. "Terramind: Large-scale generative multimodality for earth observation." arXiv preprint arXiv:2504.11171 (2025).
> 2. Mizrahi, David, et al. "4m: Massively multimodal masked modeling." Advances in Neural Information Processing Systems 36 (2023)

---

> > ### Author Response · Authors · 2025-11-19
> >
> > ***Comment Part 2***
> >
> > > The method is presented as agnostic but performance is evaluated in a single task (yield prediction).
> >
> > Thank you for highlighting this. We agree that broader evaluation is desirable; however, we were constrained by the current lack of public datasets that feature both satellite imagery and aligned weather time series. CropNet represents the most comprehensive benchmark currently available for this domain.
> > Despite being a single dataset, CropNet ensures a diverse evaluation by covering four different crop types and varied climatic regions across the US. The significant performance gains our model achieves across all these sub-tasks provide strong evidence of the transferability of our task-agnostic pretraining. We hope our results will encourage the community to curate further datasets for this promising multimodal direction.
> >
> > > Establishing model performance across several pretraining sets (ideally combinations) and downstream tasks is important to assess the generalization of the method.
> >
> > Thank you for this point. To ensure robust generalization, we specifically designed our pretraining dataset, TerraMesh [3], as a combination of two major existing datasets: MajorTOM-Core and SSL4EO-S12. This exposes the model to a diverse mixture of global distributions during training. For what concern the downstream application, please refer to the point above.
> >
> >
> > ***Reference***
> >
> > 3. Blumenstiel, Benedikt, et al. "Terramesh: A planetary mosaic of multimodal earth observation data." Proceedings of the Computer Vision and Pattern Recognition Conference. 2025.

---

### Official Review · Reviewer_suZX · 2025-11-01

**Soundness:** 2
**Presentation:** 2
**Contribution:** 2
**Rating:** 2
**Confidence:** 4

**Summary:**

This paper proposes a task-agnostic framework for multimodal fusion between time series and single-timestamp images. The approach explores deterministic and learned quantization strategies for time series and employs a masked correlation learning objective to align discrete image and time series tokens within a shared representation space. Applied to the Earth observation domain, the model is pretrained to generate global temperature profiles from satellite imagery and evaluated through counterfactual experiments. The pretrained model reportedly outperforms task-specific fusion and baseline methods across downstream tasks such as crop yield prediction.

**Strengths:**

- The idea of unifying time series and imagery into a shared latent space is interesting and potentially useful for Earth observation tasks.

- The technical components, such as masked correlation learning and modality alignment, are reasonable choices.

**Weaknesses:**

- The **motivation** for generating global temperature profiles is weak — “why we need global temperature profiles?” is not convincingly explained. Can't we see the date and location information of satellite imagery and check the temperature?

- The role of **quantization** is unclear; the paper does not adequately justify why time series need to be quantized for cross-modal alignment.

- The “quantizing time series” section is confusing and lacks logical flow between different methods.

- In Section 3.2, the inclusion of a conditional diffusion decoder (Dφ) is not well motivated — it’s unclear why generation is needed in this setup.

- The loss design (cross-entropy) in Section 3.3 seems inappropriate for the problem; an MSE-based objective between Y and Y_target may be more natural.

- The paper evaluates only one model (TerraMesh), which is insufficient to demonstrate generality.

- Downstream evaluation relies solely on the CropNet dataset, limiting the evidence for broad applicability.

- The comment in Section 5.3 about the asymmetric confidence interval (“larger positive bound”) is confusing and even suggests degraded performance.

- Overall, the paper reads more like a technical report than a mature ICLR submission — the writing and motivation need substantial refinement.

**Questions:**

Please refer to Weaknesses.

While the concept of fusing temporal and spatial modalities is promising, the paper lacks clarity, strong motivation, and sufficient empirical validation. The scope of experiments and explanations do not meet the standard expected at ICLR.

---

> ### Author Response · Authors · 2025-11-19
>
> We thank the reviewer for the constructive feedback and helpful suggestions. Please find our detailed responses to each of your concerns below.
>
> ***Comment Part 1***
>
>  > The motivation for generating global temperature profiles is weak — “why we need global temperature profiles?” is not convincingly explained. Can't we see the date and location information of satellite imagery and check the temperature?
>
> We would like to clarify that the primary objective of our paper is not to propose a new method for temperature forecasting. Rather, our central focus is on developing a **representation learning framework that effectively fuses time series and satellite imagery** by learning the correlation between them in a unified representation space. The generation of global temperature profiles is presented as a key experiment to rigorously **probe and demonstrate the robustness of this learned correlation**. This capability is an emergent property of our successful pretraining, not the primary application. By showing that our model can generate spatially and temporally consistent temperature profiles, we provide strong evidence that the learned latent space is meaningful and captures the physical relationships between the modalities. In essence, this experiment serves to answer the question: 'How can we be sure the learned cross-modal alignment is non-trivial and consistent?'
>
> > In Section 3.2, the inclusion of a conditional diffusion decoder (Dφ) is not well motivated — it’s unclear why generation is needed in this setup.
>
> We follow state-of-the-art models like TerraMind [1] in the use of diffusion decoders. Side experiments show that without diffusion, resolving man-made structures from quantized tokens becomes challenging due to the significant compression bottleneck.
>
> > 1) The “quantizing time series” section is confusing and lacks logical flow between different methods.
> 2) The role of quantization is unclear; the paper does not adequately justify why time series need to be quantized for cross-modal alignment.
> 3) The loss design (cross-entropy) in Section 3.3 seems inappropriate for the problem; an MSE-based objective between Y and Y_target may be more natural.
>
> Quantization strongly benefits correlation learning across modalities with significantly different representations e.g., sequence data, image data, timeseries sensor data [2].
> Our primary goal is to create a unified representation space where both time series and imagery can be correlated effectively. A common approach might be to regress continuous time series values from image embeddings using a loss function like MSE. However, our approach is informed by challenges encountered in recent large-scale multimodal systems. As observed in the development of models like **4M** [2] and **TerraMind** [1], directly regressing continuous values across different modalities often leads to training instabilities and convergence difficulties, especially when trying to balance losses from heterogeneous data sources.
> Inspired by these findings, we made the principled decision to discretize the time series into tokens. This **transforms the cross-modal fusion problem from a regression task into a classification task** (predicting the correct token). This formulation allows us to leverage a stable and highly scalable masked modeling objective with a cross-entropy loss, a paradigm that has proven successful for learning powerful joint representations in the cited works.
>
>
> > The paper evaluates only one model (TerraMesh), which is insufficient to demonstrate generality.
>
>  We appreciate the reviewer’s feedback. To clarify, our contribution is the task-agnostic fusion framework itself, which is designed to learn a shared representation for time series and imagery. TerraMesh is the name of the large-scale, multimodal Earth Observation dataset that we use to instantiate and rigorously evaluate our framework.
> Regarding the central point of generality, we argue that the core architectural and methodological components of our framework are inherently domain-agnostic.
> We choose the EO domain and the TerraMesh dataset precisely because of their complexity, scale and real-world relevance, making them an excellent and challenging testbed to demonstrate our method’s effectiveness. While we agree that demonstrating performance in other domains (such as healthcare or manufacturing) would further strengthen our claims of generality, and this is an exciting avenue for future work, we believe that our comprehensive experiments on scientific sensor data from the EO dataset provide a strong proof-of-concept for the proposed fusion methodology.
>
> ***References:***
>
> 1. Jakubik, Johannes, et al. "Terramind: Large-scale generative multimodality for earth observation." arXiv preprint arXiv:2504.11171 (2025).
> 2. Mizrahi, David, et al. "4m: Massively multimodal masked modeling." Advances in Neural Information Processing Systems 36 (2023): 58363-58408.

---

> > ### Author Response · Authors · 2025-11-19
> >
> > ***Comment Part 2***
> >
> > > Downstream evaluation relies solely on the CropNet dataset, limiting the evidence for broad applicability.
> >
> > Thank you for this very relevant point. We agree that evaluating on a wider variety of downstream datasets would be ideal for demonstrating broad applicability.
> > Our choice to focus on the CropNet dataset was driven by the current landscape of publicly available benchmarks in the Earth Observation domain. To the best of our knowledge, CropNet is the most prominent and well-established large-scale dataset that specifically combines the two modalities central to our paper: satellite imagery and corresponding weather time series. The scarcity of such multimodal datasets is a known challenge in the field.
> > However, we believe our strong performance on CropNet is significant. It is a complex benchmark comprising four distinct crop yield prediction tasks (corn, cotton, soybeans, and winter wheat) across diverse geographical regions. By demonstrating substantial improvements across all four tasks, we provide robust evidence that our task-agnostic pretraining learns transferable and meaningful representations.
> > Furthermore, we position our work as a foundational contribution. By proposing a successful and generalizable framework for fusing these specific modalities, we hope to not only provide a strong baseline for future work but also to **lead  the development and release of new multimodal datasets**, now that a viable method for their use has been demonstrated.
> >
> > > The comment in Section 5.3 about the asymmetric confidence interval (“larger positive bound”) is confusing and even suggests degraded performance.
> >
> > Thank you for the excellent question. We realize our explanation of the asymmetric confidence interval could be confusing, and we appreciate the opportunity to clarify our finding, which in fact points to the model’s robustness.
> > The primary conclusion of the counterfactual analysis in Section 5.3 is that the 95% of the confidence interval for the change in loss, [-0.1580, 4.3095], contains zero. **This is the key result, as it means we find no statistically significant evidence that our model relies on geolocation data for its prediction.
> > The model correctly learns to generate temperature profiles from the visual  features of the image itself.**
> > Our comment on the 'larger positive bound' was intended as a secondary point about the model's robustness. The asymmetry suggests that in the hypothetical scenario where the model was forced to use this auxiliary geo-location data, providing incorrect information would be more likely to increase the error (a positive delta) than decrease it. Therefore, this observation does not suggest degraded performance. **On the contrary, it reinforces that the model has learned the correct correlations from the visual data, and it is robust enough to ignore potentially confounding 'shortcut' information from the geo-location (zero is in the interval)**. We will revise Section 5.3 to remove this confusing secondary point and state our main conclusion more directly and clearly, emphasizing that the results demonstrate the model's robustness.
> >
> > > Overall, the paper reads more like a technical report than a mature ICLR submission — the writing and motivation need substantial refinement.
> >
> > Thank you for the feedback. We will revise the manuscript. Our revision will specifically focus on three key areas: (1) providing a stronger motivation for our quantization paradigm and its role in facilitating representation learning; (2) detailing the architectural necessity of the image diffusion decoder for our generative goals; and (3) polishing Section 5.3 to clearly articulate our findings on gradient sensitivity and robustness.

---

### Official Review · Reviewer_eT7M · 2025-11-02

**Soundness:** 3
**Presentation:** 3
**Contribution:** 3
**Rating:** 8
**Confidence:** 4

**Summary:**

The paper presents a task-agnostic framework for multimodal fusion of time series and single-timestamp images, enabling time-series generation from images. The approach quantizes both modalities into discrete tokens using deterministic quantile-based and learned Finite Scalar Quantization methods and then learning masked correlation objectives to align their latent spaces. This unified token-based representation allows bidirectional modality translation and self-supervised pretraining. The technique is applied to Earth Observation by aligning optical satellite imagery (TerraMesh dataset) with meteorological time series (NOAA GFS) and demonstrates that it can generate consistent global temperature profiles from satellite images without explicit geolocation data.

**Strengths:**

The central idea of the paper is discretizing both time series and images into a shared token space is impactful. The approach establishes a task-agnostic, generative, and interpretable framework. Introducing Finite Scalar Quantization (FSQ) for time series is a good technical contribution. FSQ provides a stable, computationally efficient way to discretize long-tailed time series distributions. Treating tokens from different modalities as mutual predictive targets lead task-agnostic learning principle which makes the framework scalable to diverse downstream tasks.

By experiments, the authors confirm that the model learns visual-temporal correlations rather than relying on positional cues. In downstream tasks (e.g., crop yield prediction using CropNet), the pretrained model outperforms task-specific fusion and baseline models, achieving better performance compared to conventional baselines. The paper also includes gradient-based sensitivity analyses, showing that modality-specific gradients reveal robustness and identify spatial regions with weak predictive signals.

The concept of discrete multimodal alignment is not entirely unprecedented, many generative models use token alignment, though the innovation here lies in extending it to temporal signals. Hence, the novelty is incremental but well-adapted to EO and time-series domains.

**Weaknesses:**

The masked correlation objective lacks a formal justification or ablation contrasting it with contrastive or mutual information-based alternatives.
While FSQ is good to use, the paper does not thoroughly analyze token efficiency versus representation quality. This limits understanding of scalability.
The autoregressive generation may conflate spatial priors with temporal correlations. No explicit temporal grounding or causal validation is included.
There is no computational efficiency analysis e.g., inference cost, quantization overhead, scaling limits.
While quantization improves cross-modal alignment but it may lose fine-grained temporal information which may be useful in analyzing high-frequency dynamics.

**Questions:**

As in limitations

---

> ### Author Response · Authors · 2025-11-19
>
> We are very **grateful for the positive assessment** and for providing such thoughtful, constructive feedback. Particularly, we are happy that the reviewer acknowledged the innovation of extending the discrete multimodal alignment to temporal signals.
>
> > The masked correlation objective lacks a formal justification or ablation contrasting it with contrastive or mutual information-based alternatives.
>
> We appreciate this thoughtful comment. Our decision to employ a masked correlation objective over contrastive or mutual information-based alternatives was a principled design choice driven by our primary goal: dense multimodal representation learning. Unlike contrastive approaches, which primarily focus on global discriminability between samples (instance discrimination), masked modeling forces the network to learn fine-grained, structural dependencies between modalities to reconstruct missing information. This choice was essential for our goal of creating a unified representation space that captures detailed spatial-temporal dynamics rather than just high-level alignment. We will revise the Methods section to formally articulate this justification, clarifying that our choice was dictated by the need for dense, structural representation learning rather than simple global alignment.
>
> > On the Analysis of Token Efficiency vs. Representation Quality
>
> We thank the reviewer for this comment regarding scalability and the trade-off between token efficiency and representation quality.
> We would like to highlight that we have performed a preliminary analysis of this trade-off in Appendix C.3.  We explicitly analyzed the impact of codebook utilization on quality for the FSQ method. As shown in Table 4, reducing the number of utilized tokens from 313 to 42 results in an increase in reconstruction error (MSE rises from 0.0023 to 0.0102). However, even at low token counts (42 tokens), the $R^2$ remains very high (0.9999), suggesting that the model can maintain strong structural representation even with high compression rate.
>
> > The autoregressive generation may conflate spatial priors with temporal correlations. No explicit temporal grounding or causal validation is included.
>
> We argue that our counterfactual analysis in Section 5.3 serves as an explicit form of causal validation against this issue. In that experiment, we swapped geo-locations and found no significant change in reconstruction loss. This demonstrates that the model is not relying on a simple spatial shortcut ('where it is'). Therefore, to generate a temporally consistent profile (e.g., with the correct diurnal cycle phase), the model must be inferring temporal cues directly from the visual data itself, such as lighting, shadow angles, or atmospheric conditions present in the satellite image ('when it is').
>
> > There is no computational efficiency analysis e.g., inference cost, quantization overhead, scaling limits.
>
> We observed marginal overhead from quantization processing 211.29 files per second in training mode. The inference speeds are accordingly higher processing over 300 files (timeseries) per second using 4 NVIDIA A100-SXM4-80GB in DDP. We fully avoid any inference costs for the non-learned quantization using quantile-based methods.
>
> > While quantization improves cross-modal alignment, it may lose fine-grained temporal information which may be useful in analyzing high-frequency dynamics.
>
> We agree with the reviewer, the benefit of the strong cross-modal representation learning based on quantization comes at the cost of potentially losing fine-grained details. However, we want to point out the strong reconstruction capabilities with close to perfect R2 scores.

---

### Note · Program_Chairs · 2026-01-17
**Submission Desk Rejected by Program Chairs**

The following references in this submission do not refer to real documents and/or have major errors in bibliographic information:

 M. Ashfaq et al. Integrating satellite imagery, climate time-series, and soil data for wheat yield forecasting in Pakistan. Agricultural and Forest Meteorology, 2025.